# ProbMoE: Differentiable Probabilistic Routing for Mixture-of-Experts

**Heng Zhao** [1]   **Zilei Shao** [2]   **Guy Van den Broeck** [2]   **Zhe Zeng** [1]

## Abstract

Mixture-of-Experts (MoE) models scale by activating only a small subset of experts per token. However, training such models remains challenging because top-$k$ routing is discrete and non-differentiable, requiring gradient estimators for expert selection whose design remains a central open problem. We introduce ProbMoE, a probabilistic routing framework that models expert selection as a distribution over cardinality-constrained expert subsets and formulates routing as probabilistic inference in this discrete subset space. We first propose ProbMoE Exact-$k$ routing, which samples $k$-expert subsets in the forward pass, and the backward pass uses gradients through each expert's exact marginal probability as a tractable surrogate for the true gradient. ProbMoE naturally generalizes to a dynamic-$k$ routing setting, where both training and inference constrain the routing cardinality to the same predefined range, allowing adaptive expert allocation per token. Across benchmarks and model backbones, ProbMoE Exact-$k$ achieves strong performance compared to competitive baselines, with improved expert utilization and routing diversity; ProbMoE Dynamic-$k$ achieves comparable performance with fewer activated experts. Code is available at: https://github.com/HengHugoZhao/ProbMoE.git

## 1. Introduction

Mixture-of-Experts (MoE) architectures have emerged as a central strategy for scaling large language models while keeping computational costs manageable (Shazeer et al., 2017; Fedus et al., 2022). By selecting only a small subset of experts for each token, MoE models achieve sublinear growth in FLOPs and allow the total parameter count to substantially exceed the active compute budget (Du et al., 2022; Liu et al., 2023; Jiang et al., 2024).

However, training MoE models remains challenging because routing relies on a hard top-$k$ operator that is discrete and non-differentiable, making it incompatible with standard gradient-based updates. In standard MoE architectures, the router computes expert scores by applying a softmax operation on expert logits and then selects the top-$k$ experts per token. To bypass differentiating the top-$k$ operator, common training procedures ignore its dependence on the router logits and propagate gradients only through the softmax probabilities (Shazeer et al., 2017; Lepikhin et al., 2021; Rajbhandari et al., 2022). As a result, the router receives limited learning signal about alternative expert subsets beyond the deterministically selected top-$k$ experts, which can lead to highly concentrated routing distributions, poor expert utilization, and unstable training dynamics (Lewis et al., 2021; Zuo et al., 2022; Clark et al., 2022).

These challenges call for a principled routing mechanism. Instead of treating routing as a deterministic operator with heuristic gradient approximations, we propose to cast expert routing as a probabilistic inference problem: for each token, the selection scores of each expert induce a distribution over expert subsets, and selecting $k$ experts corresponds to probabilistic inference under a cardinality constraint. From this perspective, training aims to optimize the expected loss under the expert subset distribution, which requires differentiating with respect to the parameters of a discrete, cardinality-constrained subset distribution.

Although the space of expert subsets is combinatorial, our proposed probabilistic formulation of MoE routing admits tractable inference. To optimize this formulation, we adopt SIMPLE (Ahmed et al., 2023), a gradient estimator for cardinality-constrained subset distributions. Our proposed framework ProbMoE uses SIMPLE to sample valid $k$-subsets of experts in the forward pass and differentiate through the induced expert-selection marginals in the backward pass, yielding routing gradients that capture how changes in the selection probability of each expert affect the expected loss. It enables stochastic exploration of expert subsets with informative router updates, while preserving

[1]Department of Computer Science, University of Virginia, Charlottesville, USA [2]Department of Computer Science, University of California, Los Angeles, Los Angeles, USA. Correspondence to: Heng Zhao <qgg5se@virginia.edu>.

*Proceedings of the 43rd International Conference on Machine Learning*, Seoul, South Korea. PMLR 306, 2026. Copyright 2026 by the author(s).

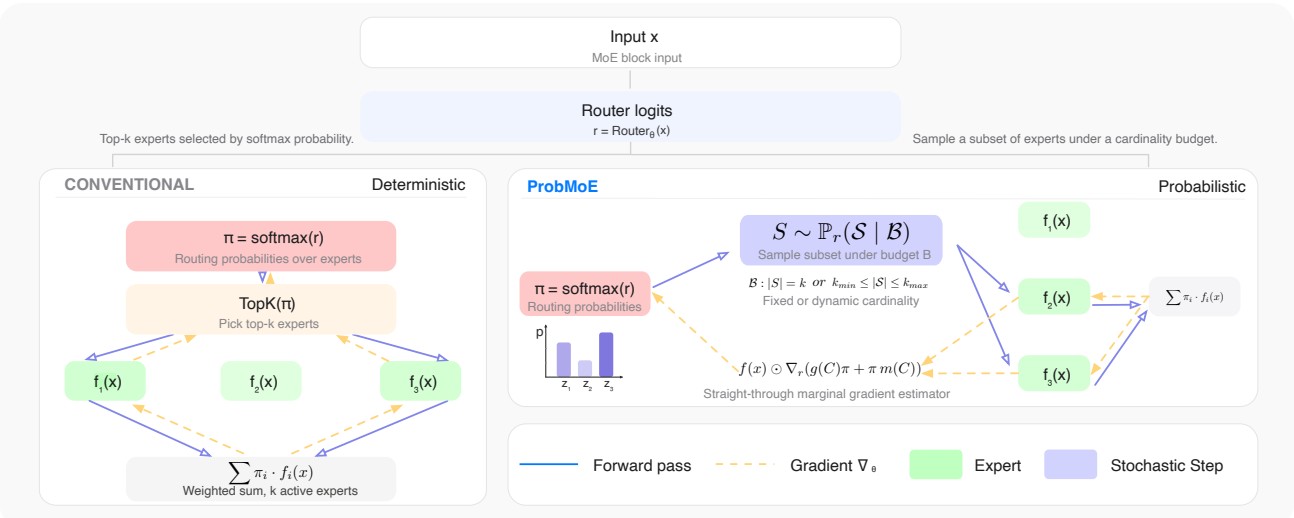

*Figure 1.* **Comparison of conventional Top-$k$ training and ProbMoE training. Left: Conventional** MoE applies a deterministic top-$k$ operator to the softmax routing probabilities for expert selection, while propagating gradients only through these probabilities. **Right: ProbMoE** models expert routing as probabilistic inference over discrete expert subsets. ProbMoE samples an expert subset $S$ from a cardinality-constrained distribution. Let $z$ denote the binary mask of the sampled subset and let $m$ denote the corresponding expert-selection marginals. Gradients are propagated through the straight-through mask $g = \text{stopgrad}(z - m) + m$, which is combined with the softmax routing probabilities to form the final routing weights. This yields informative router gradients while preserving sparse expert execution. ProbMoE Dynamic-$k$ allows the subset size to vary within a range, with ProbMoE Exact-$k$ recovered as a special case.

sparse expert execution. Moreover, ProbMoE is agnostic to the underlying MoE architecture and can be flexibly integrated into the training of existing MoE models.

Alongside improving fixed-$k$ routing, another ongoing direction in MoE research is dynamic expert allocation, where the number of activated experts can vary across tokens. It is a compelling setting because it allows the model to adapt computation to token- or task-level complexity. Such flexibility can improve efficiency and better match the computational needs of language modeling than a fixed-$k$ setting. However, dynamic-$k$ routing poses a distinct challenge: modeling the choice of expert subset size $k$. Methods designed for exact-$k$ routing assume a fixed subset size and therefore cannot be directly applied to variable cardinality selection, where the router must reason over subsets of different sizes.

To this end, our proposed probabilistic framework naturally unifies exact-$k$ and dynamic-$k$ routing in a principled way, as shown in Figure 1. ProbMoE Exact-$k$ conditions the subset distribution on a fixed cardinality, whereas ProbMoE Dynamic-$k$ conditions the same distribution on a range of feasible cardinalities. Unlike prior adaptive expert allocation methods that rely on thresholding, null experts, or other heuristic gating rules (Yue et al., 2025; Jin et al., 2025a;b), ProbMoE Dynamic-$k$ considers a normalized distribution over feasible expert subsets and enables effective training with tractable marginal-based router gradients.

**Contributions** We summarize our main contributions as follows: (1) we propose to cast MoE routing as probabilistic

inference problems over cardinality-constrained expert subsets; (2) we adopt the gradient estimator SIMPLE in MoE routing to pair probabilistic subset sampling with marginal-based router gradients; (3) we introduce a range-constrained routing variant for dynamic-$k$ MoE routing that jointly infers expert identities and routing cardinality for each token; (4) in empirical evaluations, we show that ProbMoE Exact-$k$ improves expert utilization and routing diversity; (5) in dynamic $k$ setting, experiment results show that ProbMoE Dynamic-$k$ achieves competitive performance with fewer activated experts on average.

## 2. Related Work

**Routing Optimization and Differentiability.** Standard Top-$k$ routing is non-differentiable, motivating several strategies to stabilize training. One stream of work modifies gradient propagation without changing the discrete nature of the forward pass, employing techniques such as dense backpropagation (Panda et al., 2025; Yao et al., 2026) or learned sparse gradient selection (Liu et al., 2023). Another line of work relaxes the discrete constraint, using continuous activations to approximate routing decisions (Wang et al., 2025). Although effective for improving optimization stability, these methods do not explicitly model the distribution over feasible expert subsets, and therefore provide limited mechanisms for exploring alternative expert combinations during training. Prior routers have also introduced stochastic elements, such as noise injection (Shazeer et al., 2017;

Shabgahi et al., 2026) or randomized tie-breaking (Lepikhin et al., 2021; Fedus et al., 2022). However, they primarily treat routing as a deterministic operator augmented with heuristic randomization. To our knowledge, no prior work formulates MoE routing as probabilistic inference over discrete expert subsets, using stochastic subset selection as a principled mechanism for exploration during training.

**Adaptive Expert Allocation** A parallel line of work moves beyond fixed-capacity routing to explore adaptive allocation, where the number of active experts varies by token difficulty. Approaches such as DA-MoE (Aghdam et al., 2024), DynMoE (Guo et al., 2025), and AdaMOE (Zeng et al., 2024) trade off model capacity and cost by adjusting expert counts via learned thresholds or null routers. While these and related methods achieve dynamic sparsity (Yue et al., 2025; Jin et al., 2025a;b), they rely on heuristic gating mechanisms rather than a unified probabilistic formulation. Consequently, they lack the ability to perform differentiable selection while maintaining normalization over the combinatorial space of expert subsets.

**Probabilistic Sampling** Treating the MoE router as a probabilistic layer requires differentiable sampling from discrete distributions, a problem studied extensively outside the MoE context. Common relaxations such as the Concrete and Gumbel-Softmax distributions (Maddison et al., 2017; Jang et al., 2017) enable gradient-based optimization but often suffer from high variance or bias. In contrast, Ahmed et al. (2023) formulates subset selection as probabilistic inference under explicit cardinality constraints and the proposed gradient estimator SIMPLE has shown to be effective in various learning settings (Qian et al., 2024; Shukla et al., 2023). We leverage this perspective to view the MoE router not as a deterministic switch but a probabilistic layer with tractable normalization and marginal probabilities.

## 3. Preliminaries

In this section, we review the standard formulation of MoE routing and formalize why top-$k$ expert selection poses fundamental challenges for gradient-based optimization, pointing toward probabilistic formulations of expert selection.

We consider a MoE layer with $N$ experts indexed by $[N] \triangleq \{1, \dots, N\}$. Given the hidden state of an input token $x \in \mathbb{R}^d$, the router produces a vector of logits $r = \text{Router}_\theta(x) \in \mathbb{R}^N$, where $\theta$ denotes the router parameters. Each expert is a function $f_i : \mathbb{R}^d \to \mathbb{R}^d$ indexed by $i \in [N]$.

The router logits $r$ are converted into soft routing weights using the softmax function:

$$\pi_i = \frac{\exp(r_i)}{\sum_{j=1}^N \exp(r_j)}, \qquad \forall i \in [N].$$

A sparse MoE layer then selects a subset of experts using the top-$k$ operator. Let $S_{\text{top-}k}(r) = \text{TopK}(r)$ denote the set of indices corresponding to the $k$ largest router logits. For any fixed subset $S \subseteq [N]$ with $|S| = k$, define the MoE output associated with selecting $S$ as

$$y_S(x; r) \triangleq \sum_{j \in S} \pi_j f_j(x). \tag{1}$$

The standard top-$k$ routed output can therefore be written as

$$y(x; r) = \sum_{\substack{S \subseteq [N] \\ |S|=k}} \mathbb{I}[S = S_{\text{top-}k}(r)] \; y_S(x; r).$$

Differentiating this expression with respect to a router logit $r_i$ reveals two distinct paths:

$$\frac{\partial y(x; r)}{\partial r_i} = \underbrace{\sum_{\substack{S \subseteq [N] \\ |S|=k}} \mathbb{I}[S = S_{\text{top-}k}(r)] \frac{\partial y_S(x; r)}{\partial r_i}}_{\text{softmax-weight path}}$$
$$+ \underbrace{\sum_{\substack{S \subseteq [N] \\ |S|=k}} y_S(x; r) \frac{\partial \mathbb{I}[S = S_{\text{top-}k}(r)]}{\partial r_i}}_{\text{discrete-selection path}}. \tag{2}$$

The first term in equation (2) is differentiable because the selected subset is treated as fixed and gradients flow only through the softmax routing weights inside $y_S(x; r)$. In contrast, the second term contains the derivative of the subset-selection indicator $\mathbb{I}[S = S_{\text{top-}k}(r)]$. This indicator is a discrete, piecewise-constant function of the router logits: it remains unchanged within each selection region and changes discontinuously at ranking boundaries. Therefore, this discrete selection path is non-differentiable under standard backpropagation.

Conventional MoE training effectively drops the second term in Equation (2) by treating $S_{\text{top-}k}(r)$ as fixed during the backward pass. Under this approximation, gradients with respect to $r$ are computed only through the softmax routing weights assigned to the selected experts, giving

$$\frac{\partial \mathcal{L}}{\partial r_i} = \sum_{j \in S_{\text{top-}k}} \left\langle \frac{\partial \mathcal{L}}{\partial y}, f_j(x) \right\rangle \frac{\partial \pi_j}{\partial r_i}.$$

The difficulty of training under top-$k$ routing stems from the discrete nature of expert subset selection. Conventional training provides router gradients by modifying how gradients are propagated through the selected experts. However, the resulting learning signal remains tied to a deterministic top-$k$ decision, providing limited supervision for the discrete selection mechanism itself. As a result, alternative expert subsets receive limited exploration during training, which can reinforce concentrated routing patterns and worsen expert under-utilization. Prior approaches partially mitigate

this issue by using dense straight-through estimator to provide proxies for gradients for the discrete selection (Lewis et al., 2021; Panda et al., 2025; Yao et al., 2026), but these gradients are still defined over individual expert scores rather than over the structured subset selection process. This motivates a formulation that treats routing in a principled way as probabilistic inference over expert subsets.

# 4. ProbMoE for Exact-$k$ routing

We begin by modeling MoE routing as a *probabilistic latent layer*. In a standard MoE layer, each token is routed to an expert subset, but top-$k$ routing selects this subset deterministically and is non-differentiable with respect to the router logits. Rather than treating expert selection as a fixed discrete decision in the backward pass, ProbMoE considers the distribution over feasible expert subsets and trains the router by optimizing the expected loss under this distribution, defined as below

$$\mathcal{J}(\theta) = \mathbb{E}_{S \sim \mathbb{P}_r(\cdot \,||S|=k)} \left[ \mathcal{L}\big(y_S(x; r)\big) \right], \qquad (3)$$

where $\mathbb{P}_r(\cdot \mid |S| = k)$ is the router-induced distribution over feasible subsets $S \subseteq [N]$ with $|S| = k$, and $\mathcal{L}\big(y_S(x; r)\big)$ is the point-wise task loss induced by the MoE output when subset $S$ is selected. This objective optimizes the router over a distribution of feasible expert subsets, allowing alternative expert combinations to influence training.

**Exact-$k$ subset distribution** We formally define the router-induced distribution $\mathbb{P}_r(\cdot \mid |S| = k)$ used in Equation (3). For each expert $i$, let $p_i = \sigma(r_i)$ denote its Bernoulli selection probability, where $\sigma(\cdot)$ is the sigmoid function. These Bernoulli variables induce an unconstrained product distribution over expert subsets. To model exact-$k$ routing, we condition this distribution on the constraint that exactly $k$ experts are selected. For any subset $S \subseteq [N]$ with $|S| = k$, the probability of this subset is

$$\mathbb{P}_r(S \mid |S| = k) = \frac{1}{Z_k} \prod_{j \in S} p_j \prod_{j \notin S} (1 - p_j),$$

where the normalizing constant is

$$Z_k = \sum_{S \subseteq [N], |S|=k} \prod_{j \in S} p_j \prod_{j \notin S} (1 - p_j). \qquad (4)$$

**Theorem 4.1** (Ahmed et al. (2023)). *The normalizing constant $Z_k$ in Equation* (4) *can be computed exactly in time $\mathcal{O}(Nk)$. Moreover, it can be computed in a vectorized complexity $\mathcal{O}(\log N \log k)$ assuming perfect parallelization.*

Thus, ProbMoE can define and normalize a distribution over exact-$k$ expert subsets while remaining tractable.

## 4.1. Forward pass

ProbMoE samples an expert subset from the aforementioned exact-$k$ distribution in the forward pass:

$$S \sim \mathbb{P}_r(S \mid |S| = k).$$

The MoE layer then computes the subset-specific output in Equation (1) using the sampled subset $S$. Since $|S| = k$, each forward pass still evaluates only $k$ experts per token, preserving the sparse execution pattern of standard MoE routing. Unlike deterministic top-$k$ routing, which always selects the experts with the largest router scores, this stochastic forward pass can explore alternative expert combinations. As a result, high probability alternatives to the top-$k$ set can be explored during training, preventing the same expert combination from being reinforced repeatedly while other nearly competitive subsets remain under-trained.

At inference time, ProbMoE selects the Maximum-A-Posteriori (MAP) set, making the inference cost of ProbMoE Exact-$k$ same as the conventional MoE routing.

## 4.2. Backward pass

Having defined the expected loss objective in Equation (3), we now describe how ProbMoE optimizes it in practice. Ideally, updating the router would require differentiating the expectation with respect to the router parameters:

$$\nabla_\theta \mathcal{J}(\theta) = \nabla_\theta \mathbb{E}_{S \sim \mathbb{P}_r(\cdot \,||S|=k)} \left[ \mathcal{L}\big(y_S(x; r)\big) \right]. \qquad (5)$$

However, the sampled mask is discrete, so gradients cannot be propagated directly through the expert selection decision under standard backpropagation. Therefore, the backward pass requires a differentiable proxy that captures how changes in the router logits affect the sampled subset.

ProbMoE addresses this challenge by using conditional marginals, as a surrogate for the discrete sample in the backward pass as in SIMPLE (Ahmed et al., 2023). For each expert, the marginal measures its probability of being selected under the full exact-$k$ subset distribution. Thus, the marginals provide a continuous, differentiable summary of the discrete selection process. Treating the non-selection factors $(1 - p_j)$ as constants when differentiating with respect to $\log p_j$, the marginal probability of expert $j$ is

$$m_j \triangleq \mathbb{P}_r(j \in S \mid |S| = k) = \frac{\partial \log Z_k}{\partial \log p_j}. \qquad (6)$$

Since $Z_k$ is computed exactly by dynamic programming, these marginals are exact under the conditional exact-$k$ distribution and provide a tractable proxy for expert selection.

**Marginal-integrated routing weights** The marginals alone describe selection probabilities, but the MoE layer ultimately requires routing weights that determine how expert

outputs are combined. The MoE output in Equation (1) is a weighted combination of expert outputs, so the routing mechanism must determine both which experts are selected and how their contributions are weighted. Incorporating the differentiable marginals into the routing weights allows learning to influence both discrete selection decisions and the relative contribution of selected experts, while remaining consistent with the underlying stochastic subset formulation. Let the sampled subset be represented by a $k$-hot mask $z \in \{0, 1\}^N$, where $z_i = 1$ if expert $i$ is selected and $z_i = 0$ otherwise. ProbMoE combines the sampled forward mask, the marginals, and the soft routing weights through a straight-through estimator (Bengio et al., 2013):

$$w = \big( \text{stopgrad}(z - m) + m \big) \odot \pi, \qquad (7)$$

where $\text{stopgrad}(\cdot)$ blocks gradients through its argument. In the forward pass, $\text{stopgrad}(z - m) + m = z$, so the sampled mask is preserved. In the backward pass, gradients flow through both $m$ and $\pi$, enabling joint optimization of expert selection and expert weighting.

**Resulting router gradients** With the routing weights in Equation (7), the ProbMoE gradient with respect to the router logit $r_i$ decomposes as

$$\frac{\partial \mathcal{L}}{\partial r_i} = \sum_{j=1}^{N} \left\langle \frac{\partial \mathcal{L}}{\partial y}, f_j(x) \right\rangle \left( m_j \frac{\partial \pi_j}{\partial r_i} + \pi_j \frac{\partial m_j}{\partial r_i}. \right) \qquad (8)$$

The first term in Equation (8) updates the softmax weighting of expert outputs, while the second term propagates gradients through the exact marginal probabilities. Thus, ProbMoE approximates the gradient of the expected loss objective in Equation (5) by differentiating the sampled loss through the marginal integrated routing weights in Equation (7): $\nabla_\theta \mathcal{J}(\theta) \approx \nabla_\theta \mathcal{L}\big(y(x; r)\big)$. In a synthetic experiment detailed in Appendix F, we show that this estimator yields lower gradient variance than that in DenseMixer.

## 5. ProbMoE for Dynamic-$k$ routing

ProbMoE Exact-$k$ formulation in the previous section provides a differentiable way to train MoE routers when every token is assigned the same number of experts. However, different tokens may require different amounts of expert capacity depending on their ambiguity, rarity, or task-specific complexity. This motivates dynamic MoE routing, where the number of active experts is allowed to vary across tokens. Because of the probabilistic routing formulation of ProbMoE, it naturally extends beyond fixed-cardinality expert selection. Once expert routing is modeled as probabilistic inference over subsets, conditioning on a single cardinality becomes a modeling choice rather than a necessity. This observation motivates ProbMoE *Dynamic-$k$* routing, where the subset cardinality itself is treated as a discrete latent variable to be inferred jointly with expert identities.

Formally, ProbMoE Dynamic-$k$ defines a range constraint on the selected subset size, $k \in [k_{\min}, k_{\max}]$. Rather than conditioning the router induced distribution on a fixed cardinality constraint, ProbMoE Dynamic-$k$ conditions it on the broader cardinality range. The induced distribution can be factorized by first sampling a cardinality $k$ from the marginal distribution over feasible subset sizes and then sampling an expert subset conditioned on $|S| = k$. This factorization provides an efficient sampling procedure, while the resulting routing decision remains a sampled subset of experts. Therefore, ProbMoE Dynamic-$k$ can adapt the routing budget per token while retaining the same probabilistic framework.

**Range-constrained expert selection** ProbMoE Dynamic-$k$ routing extends the probabilistic subset formulation by relaxing the fixed-cardinality constraint. As in ProbMoE Exact-$k$ routing, each expert $i$ is associated with an independent Bernoulli selection probability $p_i = \sigma(r_i)$, defining an unconstrained product measure over expert subsets $S \subseteq [N]$. ProbMoE Dynamic-$k$ conditions on the constraint that the subset size lies in a predefined range, $k_{\min} \leq |S| \leq k_{\max}$. This yields the following distribution:

$$\mathbb{P}_r(S \mid k_{\min} \leq |S| \leq k_{\max}) = \frac{1}{Z^*} \prod_{j \in S} p_j \prod_{j \notin S} (1 - p_j),$$

where $Z^*$ denotes the normalization constant.

**Theorem 5.1.** *Let $Z_k$ be the normalizing constant as in Equation (4). Then the range-constrained normalizing constant, $Z^* = \sum_{k=k_{\min}}^{k_{\max}} Z_k$, can be computed in time $\mathcal{O}(Nk_{\max})$. Moreover, it admits a vectorized complexity $\mathcal{O}(\log N \log k_{\max})$ assuming perfect parallelization.*

Proof is provided in Appendix B.1. Thus, the probability mass is renormalized over all subsets whose cardinality lies in $[k_{\min}, k_{\max}]$ using the tractable normalization constant.

**Adaptive cardinality inference.** Conditioning on the range $\mathcal{B}$ also induces a conditional distribution over subset sizes,

$$\mathbb{P}_r(|S| = k \mid k_{\min} \leq |S| \leq k_{\max})$$
$$= \frac{\mathbb{P}_r(|S| = k)}{\sum_{k'=k_{\min}}^{k_{\max}} \mathbb{P}_r(|S| = k')}, \quad k \in [k_{\min}, k_{\max}]. \qquad (9)$$

allowing the model to infer how many experts to allocate to each token. During training, we first sample a cardinality $k$ from this distribution in Equation (9) and then sample an expert subset conditioned on $|S| = k$ using ProbMoE Exact-$k$ procedure. This can be interpreted as jointly inferring a token-specific routing cardinality and the corresponding expert subset within a single probabilistic model. We discuss practical choices of the $k_{\min} : k_{\max}$ range in Appendix C.

ProbMoE Dynamic-$k$ retains differentiable expert-selection marginals for the backward pass. In particular, under the range constraint, each expert's marginal probability can be

*Table 1.* **Performance comparison across benchmarks.** Top-3 are highlighted; darker colors indicate better performance. GSM is evaluated by Exact Match; Law, Translation, and Summary by LLM-as-judge; MBPP and MMLU by LM Evaluation Harness.

| Model | GSM | Law | Translation | MBPP | Summary | MMLU Overall | MMLU Stem |
|---|---|---|---|---|---|---|---|
| **OLMoE Backbone (Top-$k = 8$)** | | | | | | | |
| Base Model | 15.85 | 5.70 | 11.09 | 19.80 | 7.40 | 53.40 | 44.91 |
| Frozen Router | 44.88 | 22.50 | 28.29 | 17.80 | 36.05 | 53.97 | 46.18 |
| Conventional | 45.94 | 25.00 | 27.56 | 23.20 | 33.70 | **54.04** | **46.46** |
| DenseMixer | 47.00 | 27.90 | 30.32 | **24.40** | 37.50 | 53.95 | 46.18 |
| **ProbMoE** | **50.19** | **29.00** | **31.63** | 22.80 | **39.29** | 53.69 | 45.54 |
| **Qwen backbone (Top-$k = 4$)** | | | | | | | |
| Base Model | 38.69 | 18.20 | 16.53 | **38.84** | 28.29 | 60.85 | 52.27 |
| Frozen Router | 53.37 | 33.01 | 32.75 | 35.20 | 38.29 | **61.08** | 53.19 |
| Conventional | 53.30 | 29.50 | 30.00 | 32.80 | 39.00 | 61.03 | 53.16 |
| DefaultMoE | 51.00 | 31.20 | 24.00 | 33.20 | 38.40 | – | – |
| SparseMixer | 1.30 | 3.40 | 3.50 | 0.00 | 2.10 | – | – |
| ReMoE | 46.30 | 25.50 | 16.99 | 33.00 | 25.80 | – | – |
| DenseMixer | **54.97** | 30.75 | 33.75 | 34.00 | 41.00 | 61.03 | 52.87 |
| **ProbMoE** | 53.29 | **34.40** | **39.23** | 35.00 | **44.40** | 61.05 | **53.82** |

*Note: Baseline results are taken from the GitHub of Yao et al. (2026). The MMLU results for pretraining-based baselines are not included because reproducing these methods under the same setting would require substantially more compute than is available.*

obtained by differentiating the range-constrained normalizing constant with respect to the Bernoulli parameters. These marginals provide continuous surrogates for discrete expert selection during backpropagation, allowing router gradients to reflect the full range-constrained subset distribution.

**Proposition 5.2.** *Let $Z^*$ denote the normalizing constant over all subsets satisfying $k_{\min} \leq |S| \leq k_{\max}$. For each expert $j$, the marginal probability under the range-constrained distribution satisfies that*

$$m_j^* \triangleq \mathbb{P}_r(z_j = 1 \mid k_{\min} \leq |S| \leq k_{\max}) = \frac{\partial \log Z^*}{\partial \log p_j}.$$

We provide the detailed proof in Appendix B.2. These range-constrained marginals provide differentiable expert-selection probabilities under the dynamic-$k$ subset distribution. During training, we use them in the same straight-through routing-weight construction as in ProbMoE Exact-$k$ routing in Equation (7), replacing $m_j$ with $m_j^*$. At inference time, ProbMoE Dynamic-$k$ selects the MAP subset under the range constraint, allowing the model to choose both the expert identities and the number of active experts within $[k_{\min}, k_{\max}]$. We summarize this process in Algorithm 1.

## 6. Experimental Setup

**Models** We evaluate our method on two representative MoE architectures: OLMoE-1$B$-7$B$ (Muennighoff et al., 2025), which activates 1$B$ parameters out of 7$B$ total, and Qwen1.5-MoE-A2.7$B$ (Team, 2024), which contains 14.3$B$ parameters in total with 2.7$B$ activated at inference time. OLMoE follows a standard sparse MoE design with independent experts, consisting of 16 transformer layers, each

with 64 experts, from which exactly 8 experts are activated per token. In contrast, Qwen employs a shared-expert architecture with 24 layers, where each layer contains 60 routed experts alongside 4 shared experts. All routing methods are applied consistently within each backbone, preserving the expert parameterization and architectural design. For Qwen, ProbMoE is applied only to the routed expert subset, leaving the shared experts unchanged, ensuring a fair comparison.

**Baselines** We compare against several strong routing strategies. *Frozen Router* serves as a control baseline in which the router parameters are kept fixed during training. *Conventional* refers to the standard top-$k$ MoE routing with discrete expert selections and sparse backpropagation. *DenseMixer* uses straight-through gradient estimators and performs dense gradient propagation during training while retaining sparse top-$k$ routing at inference time. For the Qwen backbone, we include additional baselines commonly used in prior work. *DefaultMoE* provides dense gradient signals to the router by approximating unselected experts, enabling dense backpropagation for the gating mechanism while maintaining sparse expert activation (Panda et al., 2025). *SparseMixer* introduces a learned sparse backpropagation mechanism that dynamically determines which experts receive gradients, balancing efficiency and stability (Liu et al., 2023). Finally, *ReMoE* replaces discrete routing with continuous ReLU-based expert selection, enabling fully differentiable training without straight-through estimators (Wang et al., 2025). Training details and hyperparameters for reproducibility are provided in Appendix A.2.

For a fair and controlled comparison, we follow the exper-

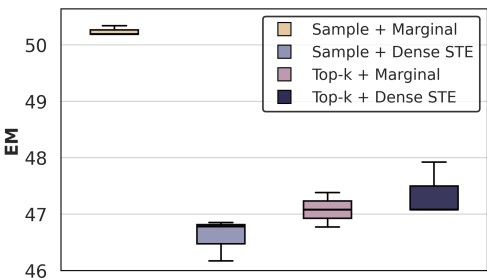

*Figure 2.* **Ablation study of forward routing and backward gradient estimation under exact-$k$ routing on OLMoE for GSM.** Box plots show exact-match (EM) accuracy, where higher is better.

imental setup of DenseMixer (Yao et al., 2026), using the same datasets, data splits, and evaluation protocols. We evaluate ProbMoE across a diverse set of tasks, including mathematical reasoning on GSM8K (Cobbe et al., 2021), legal-domain understanding, machine translation and summarization (Wang et al., 2024), and code generation. For the coding domain, we fine-tune on CodeAlpaca (Chaudhary, 2023) and evaluate on MBPP (Austin et al., 2021). Using the same CodeAlpaca fine-tuned model, we additionally report MMLU (Hendrycks et al., 2021) to assess general knowledge retention, with MMLU-Stem reported separately as a code-relevant subset. See Appendix A.3 for evaluation.

# 7. Exact-$k$ Routing Experiments

This section evaluates ProbMoE under the exact-$k$ routing setting across multiple backbones and tasks, comparing overall performance and routing behavior against baselines. As shown in Table 1, ProbMoE matches or outperforms strong baselines across backbones and tasks. In particular, ProbMoE achieves state-of-the-art results on multiple benchmarks under both OLMoE and Qwen backbones, demonstrating the effectiveness of probabilistic subset routing across diverse settings. These gains are notable given that ProbMoE preserves sparse expert execution and sparse expert-weight updates at both training and inference time, while providing dense learning signals only to the router, whereas DenseMixer introduces dense expert-side computation and gradient updates during training. Performance on GSM under the Qwen backbone exhibits a different pattern: ProbMoE does not rank among the top three methods, but remains comparable to Conventional Routing. To understand when and why ProbMoE improves performance, we next analyze routing distributions and expert utilization. These analyses show that ProbMoE induces better-calibrated routing distributions and more stable expert engagement than baseline methods thanks to the probabilistic framework.

## 7.1. Ablation Study of the Probabilistic Framework

To understand whether the gains of ProbMoE arise from stochastic expert sampling, marginal-based optimization, or

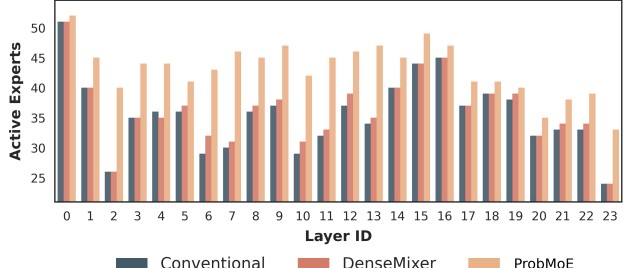

*Figure 3.* **ProbMoE Exact-$k$ activates more diverse experts across layers on Qwen than Conventional and DenseMixer routing.** We measure the minimum number of experts required to cumulatively account for $99\%$ of the routing probability mass per token, using a random subset of Translation dataset ($N = 1,000$).

their combination, we perform an ablation study where we decouple the forward routing mechanism from the backward gradient estimator. We evaluate four routing configurations on the OLMoE backbone, each fine-tuned over three independent random seeds: **Routing variants:** ProbMoE (Sample + Marginal): stochastic sampling with exact marginal gradients; DenseMixer (Top-$k$ + Dense STE): deterministic Top-$k$ with dense STE gradients; Sample + Dense STE: stochastic sampling with STE gradients; Top-$k$ + Marginal: deterministic Top-$k$ with exact marginal gradients.

As shown in Figure 2, the ProbMoE configuration achieves the highest mean Exact Match (EM) score ($50.24\%$) with low variance ($\sigma \approx 0.09$). In contrast, the other combinations exhibit clear degradation. *Sample + Dense STE* reduces performance to $46.6\%$ and substantially increases variance ($\sigma \approx 0.37$). These results indicate that the gains of ProbMoE stem from the alignment between probabilistic expert selection and marginal-based optimization.

## 7.2. Analysis of Routing Distribution diversity

To examine routing behavior beyond aggregate performance, we analyze how routing probability is distributed across experts. Prior work show that highly concentrated routing can lead to expert collapse and inefficient use of model capacity, whereas distributing probability mass more broadly encourages expert specialization and more stable training dynamics (Shazeer et al., 2017; Lepikhin et al., 2021; Zuo et al., 2022; Clark et al., 2022). Rather than focusing only on which experts are selected, we study how routing probability accumulates across the ranked experts, providing a direct measure of routing diversity and expert utilization.

Specifically, we measure how many experts are needed to account for a given fraction of the total routing probability for each token and layer. This cumulative view captures not only the most dominant experts but also the contribution of lower-ranked experts, offering a more complete picture of how broadly routing probability is distributed during inference. All statistics are computed per token and per

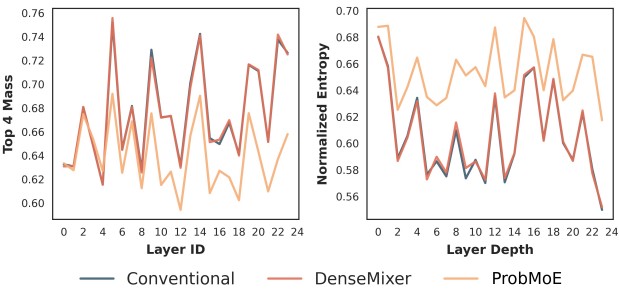

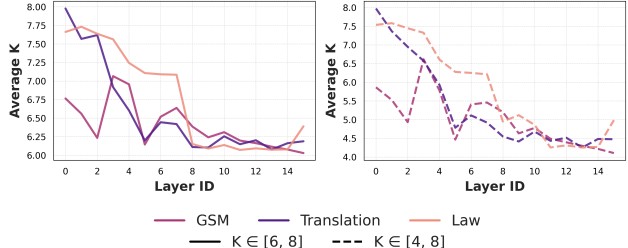

*Figure 5.* Average routing cardinality at each layer under ProbMoE Dynamic-$k$ with OLMoE backbone on different datasets.

*Figure 4.* **ProbMoE Exact-$k$ yields less concentrated and more balanced expert assignment across layers on the Qwen-MoE translation dataset.** The left panel reports the Top-4 assignment mass, defined as the fraction of assignments captured by the four most frequently used experts per layer, while the right panel shows the normalized entropy of the expert assignment distribution.

*Table 2.* Performance of Dynamic-$k$ relative to Exact-$k$. Values are reported as absolute **EM differences**, with the **average fraction of experts used** shown in parentheses.

| Dataset | OLMoE | Qwen1.5 |
|---|---|---|
| GSM | −1.82 (80.00%) | −4.29 (75.00%) |
| Law | −0.04 (84.50%) | +2.70 (75.00%) |
| Translation | +0.36 (82.00%) | +3.22 (75.00%) |

layer, and then aggregated across tokens.

Figure 3 and Figure 4 illustrate how routing probability mass is distributed across experts. Across both backbones, ProbMoE requires a larger number of experts to reach the same cumulative mass threshold $\alpha$, indicating broader routing support. This effect is especially pronounced on Qwen fine-tuned for Translation, which uses a large expert pool of 60 experts per layer while selecting only four experts per token. Under this highly competitive routing regime, ProbMoE consistently requires more experts to reach 99% cumulative probability mass across nearly all layers, particularly in deeper layers. Because small shifts in routing probability correspond to large changes in token allocation under such extreme sparsity, this pattern reflects a substantially wider routing distribution induced by ProbMoE.

To further examine how routing probability is allocated within each layer, we analyze expert utilization on Qwen using Top-4 mass and normalized entropy. Top-4 mass is computed as the fraction of total expert assignments in a layer accounted for by the four most frequently used experts, while normalized entropy measures the overall evenness of the assignment distribution. As in Figure 4, ProbMoE exhibits lower Top-4 mass and higher normalized entropy across layers than both Conventional and DenseMixer routing, indicating that expert traffic is distributed more evenly. In contrast, Conventional and DenseMixer routing produce highly peaked within-layer distributions, with a small number of experts dominating token assignments.

Taken together, these results indicate that ProbMoE exposes

a broader set of experts to routing probability over the course of training. Rather than relying on a small group of consistently dominant experts, ProbMoE distributes routing probability more broadly, allowing a wider range of experts to participate across prompts and layers. This broader routing distribution is consistent with ProbMoE's marginal-based optimization. By propagating gradients through the expert-selection marginals in Equation (6), ProbMoE provides informative learning signals to the router even when only a sparse subset of experts is executed in the forward pass. This encourages routing probabilities to be shaped by the full subset distribution rather than only the selected experts. More generally, broader expert participation is known to mitigate expert collapse and improve specialization, leading to more stable training dynamics and better use of model capacity (Wu et al., 2024; Dai et al., 2024). We therefore attribute ProbMoE's performance gains to broader expert utilization and reduced routing concentration.

## 8. Dynamic Routing Experiments

This section examines ProbMoE Dynamic-$k$ routing, which relaxes the fixed-cardinality constraint and allows the model to adapt expert allocation to token-level routing uncertainty. In Table 2, ProbMoE Dynamic-$k$ achieves performance comparable to ProbMoE Exact-$k$ while activating fewer experts, reflecting that adaptive cardinalities are learned during training. This indicates that ProbMoE Dynamic-$k$ can reduce the effective routing cost relative to ProbMoE Exact-$k$ routing without substantially sacrificing task performance.

### 8.1. Dynamic Routing vs. Token-Level Difficulty

At the dataset level, ProbMoE Dynamic-$k$ allocates systematically different amounts of expert capacity across tasks depending on their difficulty. As shown in Figure 5, Law consistently activates more experts on average than Translation, while GSM activates the fewest. This separation persists across layers, indicating that Dynamic-$k$ learns task-specific routing budgets rather than enforcing a uniform expert-usage pattern, consistent with prior observations that different tasks exhibit distinct expert-allocation profiles in MoE models (Huang et al., 2024).

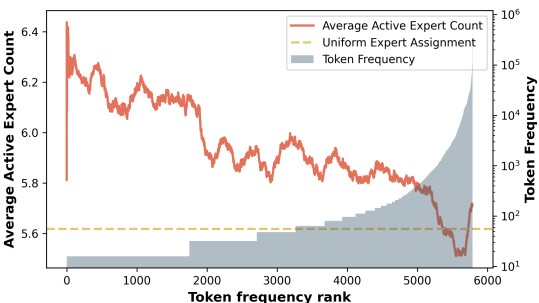

*Figure 6.* **Token frequency versus average expert activation under ProbMoE Dynamic-$k$ routing (over 656k tokens).** Tokens are ordered by increasing frequency. The solid curve shows the average number of active experts per token, while the shaded histogram (right axis, log scale) shows token frequency. Dashed line indicates the expected activation under uniform expert assignment, with rarer tokens activating more experts than frequent ones.

Under ProbMoE Dynamic-$k$, each token activates a variable-size subset of experts, allowing the model to adaptively allocate computation under an explicit cardinality constraint. We analyze the relationship between token frequency and the average number of active experts to understand how this adaptivity is expressed at the token level.

As shown in Figure 6, ProbMoE Dynamic-$k$ consistently assigns a larger number of experts to rarer tokens, while routing more conservatively for frequent tokens. This relationship is continuous rather than piecewise: average routing cardinality varies gradually across the token frequency spectrum, rather than collapsing tokens into a small number of discrete routing regimes. Qualitative inspection of representative tokens as in Table 6 reveals a clear semantic distinction. Tokens associated with higher routing cardinality include punctuation, morphological fragments, and context-sensitive symbols (e.g., :, ?, ons), whose meaning depends strongly on surrounding structure. In contrast, tokens routed with fewer experts are typically numerals or semantically concrete words. This pattern suggests that ProbMoE Dynamic-$k$ allocates additional expert capacity to tokens whose semantics are more ambiguous or composition-dependent, while effectively "compressing" the representation of frequent, stable tokens by routing them through a smaller expert set. In contrast to approaches based on continuous expert mixing, ProbMoE realizes this adaptivity through probabilistic inference over discrete expert subsets, preserving sparse execution while allowing the compute budget itself to vary (Wang et al., 2025).

### 8.2. Training–Inference Mismatch in Cardinality Usage

Table 3 examines how different training-time routing and gradient mechanisms affect inference-time expert usage when all models are evaluated with the same dynamic-$k$ MAP inference rule. We focus on GSM under the OL-MoE backbone, where conventional MoE training uses the

*Table 3.* Dynamic inference with OLMoE on GSM ($k \in [4, 8]$).

| Training Method | EM (%) | Avg. $k$ |
|---|---|---|
| ProbMoE (Dynamic-$k$) | 44.50 | 5.018 |
| DenseMixer | 38.97 | 5.292 |
| Conventional | 38.59 | 5.039 |

exact-$k$ routing with $k = 8$. Although DenseMixer and Conventional models are trained under an exact-$k$ routing assumption, evaluating them under dynamic-$k$ MAP inference reveals that their learned routing distributions favor substantially smaller expert subsets on average. Importantly, this does not imply that these models fail to utilize all $k = 8$ experts when exact-$k$ inference is enforced; rather, it reflects the concentration structure of the routing distribution learned during training. When the inference-time cardinality constraint is relaxed, MAP subset selection exposes this concentration by selecting fewer experts with high probability.

In contrast, ProbMoE is trained to explicitly model cardinality as part of the routing objective, leading to improved task performance under dynamic-$k$ inference while maintaining a comparable average routing cardinality. These results highlight a mismatch between training-time cardinality constraints and the structure of the learned routing distribution in conventional exact-$k$ training. While exact-$k$ enforces a fixed number of selected experts during training, it does not encourage the routing distribution itself to support flexible expert allocation. As analyzed in Section 7.2, exact-$k$ routing typically exhibits high utilization within the selected subset but produces highly peaked routing distributions. Under dynamic-$k$ MAP inference, this peakiness manifests as smaller inferred expert subsets, consistent with prior observations in sparse MoE models (Shazeer et al., 2017; Lepikhin et al., 2021; Clark et al., 2022).

## 9. Conclusions & Future Directions

We propose to cast MoE routing as a *probabilistic inference* problem over discrete expert subsets under explicit cardinality constraints. By leveraging SIMPLE (Ahmed et al., 2023), ProbMoE enables tractable normalization and exact expert-selection marginals, yielding informative router gradients while preserving sparse computation. Under ProbMoE Exact-$k$, this leads to improved expert utilization while preserving sparse computation. Extending this formulation to dynamic-$k$, we show that adaptive cardinality learned during training achieves performance comparable to ProbMoE Exact-$k$ while activating fewer experts on average, adapting naturally to token and task level complexity. Together, they demonstrate that modeling MoE routing probabilistically provides a principled method for expert selection. Future work includes extending ProbMoE to pre-training and system-level optimizations that fully exploit the sparsity in the dynamic setting.

## Acknowledgments

We would like to thank Feng Yao for the helpful discussion. The authors acknowledge the Research Computing at the University of Virginia. This work was funded in part by the DARPA ANSR and CODORD programs under awards FA8750-23-2-0004 and HR00112590089, and gifts from Cisco Research, Qualcomm, and Amazon. Approved for public release; distribution is unlimited.

## Impact Statement

This paper presents work whose primary goal is to advance the methodological foundations of Mixture-of-Experts routing in machine learning models. By providing a principled probabilistic framework for discrete and adaptive expert selection, our work aims to improve training stability, interpretability, and computational efficiency in large-scale neural architectures. As such, the potential societal impacts of this work are largely indirect and consistent with those of prior advances in scalable machine learning systems. Improved routing mechanisms may enable more efficient use of computational resources and facilitate the deployment of large models under constrained budgets. At the same time, as with other improvements to model capacity and efficiency, downstream applications may inherit existing risks associated with large language models, including misuse or unintended biases, which are orthogonal to the contributions of this paper. We do not identify any new ethical concerns introduced specifically by the proposed routing framework beyond those already present in the broader deployment of machine learning models.

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

# A. Methodological and Implementation Details

### A.1. DenseMixer: Dense Training-Time Routing

DenseMixer addresses the non-differentiability of top-$k$ routing by modifying gradient propagation during training (Yao et al., 2026). Rather than modeling expert subset selection directly, it introduces a straight-through approximation that alters how gradients are propagated through the routing decision. Concretely, DenseMixer augments the softmax routing gradient with an identity term on the selected experts, yielding an approximate gradient of the form:

$$\frac{\partial \mathcal{L}}{\partial r_i} \approx \sum_{j \in \mathcal{E}} \left\langle \frac{\partial \mathcal{L}}{\partial y}, f_j(x) \right\rangle \tilde{J}_{j,i}, \tag{10}$$

where

$$\tilde{J}_{j,i} = \frac{\partial \pi_j}{\partial r_i} + \mathbb{I}[j \in S_{\text{top-}k}] \, \mathbb{I}[i = j] \tag{11}$$

augments the softmax Jacobian with a straight-through term that treats the top-$k$ selection as an identity mapping on the selected coordinates during backpropagation. This allows gradients to flow to all experts during training, while inference-time routing remains unchanged.

### A.2. Training Protocol

All methods are trained using the same optimizer, learning rate schedules, batch sizes, precision settings, and routing hyperparameters as DenseMixer. For most datasets, we adopt the same number of training epochs as in the original setup. Due to the increased optimization difficulty introduced by stochastic routing, ProbMoE occasionally requires additional training steps to reach stable convergence; in these cases, we extend training to up to 8 epochs while keeping all other hyperparameters fixed.

MMLU-overall and MMLU-stem are evaluated using the fine-tuned model on the CodeAlpaca dataset.

*Table 4.* Training configurations for MoE models (full fine-tuning only). We follow the exact setup of Yao et al. (2026).

| Model | Method | Batch Size (BS) | Learning Rate (LR) |
|---|---|---|---|
| Qwen1.5-MoE | Full | $\{32, 64, 128\}$ | $\{5e{-}7, \ldots, 4e{-}5\}$ |
| OLMoE | Full | $\{64, 128, 256, 512\}$ | $\{5e{-}7, \ldots, 4e{-}5\}$ |

*Table 5.* Training and evaluation sample counts for each dataset.

| Dataset | Training Samples | Test Samples |
|---|---|---|
| GSM | 7,473 | 1,319 |
| Law | 927 | 100 |
| Translation | 11,639 | 100 |
| Summary | 19,578 | 100 |
| CodeAlpaca | 22,000 | – |
| MBPP | – | 500 |
| MMLU-Overall | – | 14,042 |
| MMLU-Stem | – | 3,153 |

### A.3. Evaluation Protocol

We evaluate each task using the following protocols.

**Exact Match.** GSM8K is evaluated using Exact Match (EM) accuracy, where a prediction is counted as correct if and only if it exactly matches the ground-truth answer string after normalization.

**LLM-as-Judge.** Law, Translation, and Summary are evaluated using GPT-4o-mini [1] as an automated judge. The judge is prompted to score each model output against the reference answer, and the resulting scores are averaged across the test set.

**LM Evaluation Harness.** MMLU and MBPP are evaluated using the EleutherAI Language Model Evaluation Harness (Gao et al., 2024)[2] under its default task configurations.

## B. Theoretical Derivations

**Theorem B.1.** *Let $Z_k$ be the normalizing constant as in Equation* (4). *Then the range-constrained normalizing constant,*

$$Z^* = \sum_{k=k_{\min}}^{k_{\max}} Z_k$$

*can be computed in time $\mathcal{O}(Nk_{\max})$. Moreover, it admits a vectorized complexity $\mathcal{O}(\log N \log k_{\max})$ assuming perfect parallelization.*

*Proof.* By definition, the Dynamic-$k$ normalizing constant sums the Bernoulli subset mass over all subsets whose cardinality lies in the band:

$$Z^* = \sum_{\substack{S \subseteq [N] \\ k_{\min} \leq |S| \leq k_{\max}}} \prod_{j \in S} p_j \prod_{j \notin S} (1 - p_j).$$

Grouping subsets by their cardinality gives

$$Z^* = \sum_{k=k_{\min}}^{k_{\max}} \sum_{\substack{S \subseteq [N] \\ |S|=k}} \prod_{j \in S} p_j \prod_{j \notin S} (1 - p_j)$$

$$= \sum_{k=k_{\min}}^{k_{\max}} Z_k.$$

Following SIMPLE (Ahmed et al., 2023), each $Z_k$ can be computed by a dynamic program. Let $A(i, k)$ be the probability of selecting exactly $k$ experts among the first $i$ experts. Then

$$A(i, k) = p_i A(i - 1, k - 1) + (1 - p_i) A(i - 1, k),$$

with boundary conditions $A(0, 0) = 1$ and $A(0, k) = 0$ for $k > 0$. After filling the table for all $i \leq N$ and $k \leq k_{\max}$, we have $A(N, k) = Z_k$. Therefore,

$$Z^* = \sum_{k=k_{\min}}^{k_{\max}} A(N, k).$$

The dynamic program has $O(Nk_{\max})$ entries, and each entry is computed in constant time, so $Z_{k_{\min}:k_{\max}}$ can be computed in $O(Nk_{\max})$ time. $\square$

**Proposition B.2.** *Let $Z^*$ denote the normalizing constant over all subsets satisfying $k_{\min} \leq |S| \leq k_{\max}$. For each expert $j$, the marginal probability under the range-constrained distribution satisfies that*

$$m_j^* \triangleq \mathbb{P}_r(z_j = 1 \mid k_{\min} \leq |S| \leq k_{\max}) = \frac{\partial \log Z^*}{\partial \log p_j}.$$

*Proof.* By definition, the range normalizing constant sums the Bernoulli subset mass over all feasible subsets:

$$Z^* = \sum_{\substack{S \subseteq [N] \\ k_{\min} \leq |S| \leq k_{\max}}} \prod_{e \in S} p_e \prod_{e \notin S} (1 - p_e).$$

---

[1] gpt-4o-mini-2024-07-18
[2] https://github.com/EleutherAI/lm-evaluation-harness

Differentiating $Z^*$ with respect to $\log p_j$, and treating the non-selection factor $(1-p_j)$ as constant in $\log p_j$ (the stop-gradient convention of SIMPLE(Ahmed et al., 2023)), only the subsets containing expert $j$ contribute. Since $\partial p_j / \partial \log p_j = p_j$, the factor $p_j$ in each such subset is reproduced, giving the unnormalized mass of feasible subsets that include $j$:

$$\frac{\partial Z^*}{\partial \log p_j} = \sum_{\substack{S \subseteq [N] \\ k_{\min} \leq |S| \leq k_{\max},\, j \in S}} \prod_{e \in S} p_e \prod_{e \notin S} (1 - p_e).$$

The conditional probability of including expert $j$ under the range constraint is the probability mass of feasible subsets that contain $j$, normalized by the total feasible mass:

$$\mathbb{P}_r(j \in S \mid k_{\min} \leq |S| \leq k_{\max}) = \frac{\sum_{\substack{S \subseteq [N] \\ k_{\min} \leq |S| \leq k_{\max},\, j \in S}} \prod_{e \in S} p_e \prod_{e \notin S}(1 - p_e)}{Z^*}.$$

Using the previous identity, the numerator is $\frac{Z^*}{\partial \log p_j}$. Therefore,

$$\mathbb{P}_r(j \in S \mid k_{\min} \leq |S| \leq k_{\max}) = \frac{1}{Z^*} \frac{\partial Z^*}{\partial \log p_j}.$$

Finally, applying the chain rule to the logarithm gives

$$\frac{\partial \log Z^*}{\partial \log p_j} = \frac{\partial \log Z^*}{\partial Z^*} \frac{\partial Z^*}{\partial \log p_j} = \frac{1}{Z^*} \frac{\partial Z^*}{\partial \log p_j}.$$

Combining the two identities yields

$$\frac{\partial \log Z^*}{\partial \log p_j} = \mathbb{P}_r(j \in S \mid k_{\min} \leq |S| \leq k_{\max}).$$

$\square$

# C. Choosing the Dynamic-$k$ Range

The range $[k_{\min}, k_{\max}]$ controls the allowable per-token compute budget for ProbMoE Dynamic-$k$ routing. The upper bound $k_{\max}$ determines the maximum number of experts that can be activated for any token, and can therefore be chosen according to the available compute budget or latency constraint. Setting a smaller $k_{\max}$ enforces a stricter upper bound on routing cost, while a larger $k_{\max}$ allows the model to allocate additional capacity to tokens with higher routing uncertainty or task-specific complexity.

The lower bound $k_{\min}$ determines the minimum number of experts that remain active for each token. In practice, this value can be selected based on stability and capacity considerations. A very small $k_{\min}$ may reduce computation, but can also make routing overly sparse and limit expert participation. A larger $k_{\min}$ ensures a minimum level of expert usage, which can be helpful when the task requires broader representation capacity or when stable routing behavior is desired.

Together, $k_{\min}$ and $k_{\max}$ define the range over which ProbMoE Dynamic-$k$ can adapt its routing cardinality. This range provides a direct way to balance adaptive expert allocation with explicit control over computational cost. In our experiments, we choose the range so that $k_{\max}$ matches the original fixed-$k$ budget of the backbone, while $k_{\min}$ allows the model to reduce the number of active experts when fewer experts are sufficient.

# D. Extended ProbMoE Dynamic-$k$ table

*Table 6.* Dynamic routing expert analysis. Examples for top/bottom 20% experts by average $k$.

| Avg. Active Experts | Examples |
|---|---|
| Top 20% | \n, :, ?, ons, eli |
| Bottom 20% | 16, 5, downward, Meat, reduce |

## E. Wall-Clock/GPU-Memory Analysis

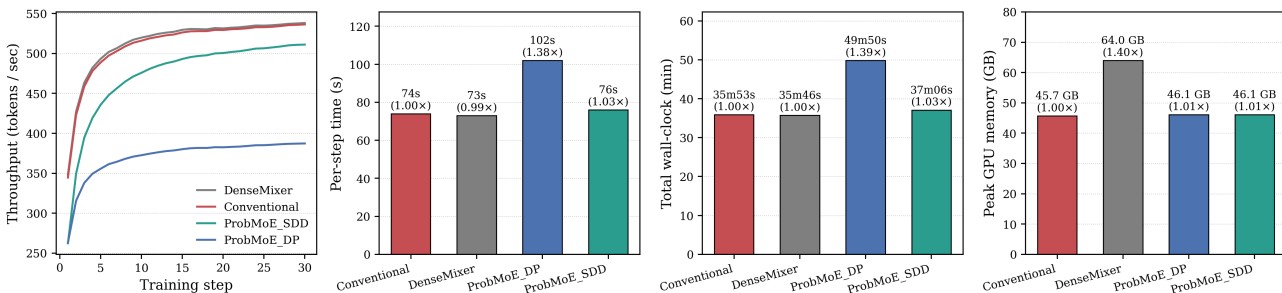

*Figure 7.* Wall-clock and memory analysis on the GSM fine-tuning task. Comparison across Conventional fine-tuning, DenseMixer, ProbMoE_DP, and ProbMoE_SDD on OLMoE-1B-7B. (a) Per-step throughput in tokens per second over 30 training steps. (b) Steady-state per-step compute time. (c) End-to-end wall-clock time. (d) Peak GPU memory per GPU. Values in parentheses denote ratios relative to Conventional.

Figure 7 reports per-step throughput, steady-state per-step compute, end-to-end training time, and peak GPU memory for the four methods on the GSM fine-tuning task. ProbMoE_SDD nearly matches Conventional fine-tuning across all three time-based metrics, with only a marginal slowdown in per-step compute and end-to-end wall-clock. ProbMoE_SDD is the optimized variant of ProbMoE_DP detailed in Algorithm 4.

In terms of peak GPU memory, ProbMoE_DP and ProbMoE_SDD closely match Conventional fine-tuning, whereas DenseMixer requires substantially more memory than any other method. DenseMixer requires a dense forward pass through all experts for every token in order to compute its straight-through gradient, dramatically increasing forward-pass FLOPs. In contrast, ProbMoE's routing optimization operates over $N$ scalar routing logits with $\mathcal{O}(N \cdot k_{\max})$ complexity, which is substantially cheaper than running full FFN computations for all inactive experts.

## F. Gradient variance comparison

*Table 7.* Comparison of aggregated bias, variance, and gradient error between ProbMoE and DenseMixer on a synthetic MoE task.

| Model | Aggregated Bias | Aggregated Variance | Aggregated Error |
|---|---|---|---|
| ProbMoE | $0.0702 \pm 0.0426$ | $0.0076 \pm 0.0110$ | $0.0967 \pm 0.0659$ |
| DenseMixer | $0.1052 \pm 0.0839$ | $0.0201 \pm 0.0427$ | $0.1349 \pm 0.0995$ |

Calculating exact ground-truth gradients on real data is intractable. In order to accurately measure an estimator's bias and variance for a comprehensive comparison, we consider a controlled setting where the true gradient is known. We carry out a synthetic experiment with a top-$K$ distribution over $n = 10$ experts and $k = 5$, yielding $C(10, 5) = 252$ possible subsets, making the exact gradient tractable in closed form. The setup simulates $T = 10$ tokens with per-token router logits to mimic real MoE behavior, and we compare DenseMixer against ProbMoE. We measure bias, variance, and average error via cosine distance, a widely adopted metric for assessing gradient variance (Ahmed et al., 2023), against the exact gradient over 10,000 samples across 10 random seeds. As shown in Table 7, ProbMoE consistently outperforms DenseMixer across all three metrics, achieving lower variance, lower bias, and lower overall gradient error, confirming that ProbMoE's marginal-based estimator provides a more accurate and stable approximation of the true gradient.

# G. ProbMoE Dynamic-$k$ Chain-DP Implementation

We summarize the overall PROBMOE Dynamic-$k$ framework in Algorithm 1, the core probabilistic sampling and marginal computation logic in Algorithm 2, and the deterministic Maximum A Posteriori (MAP) routing procedure used for inference in Algorithm 3.

---

**Algorithm 1** PROBMOE($H, k_{\min}, k_{\max}$): ProbMoE Dynamic-$k$ Routing for MoE

---

**input** Token states $H \in \mathbb{R}^{T \times d}$, router logits $r$, $k_{\min}, k_{\max}$
**output** Selected experts $I$, routing weights $w$
    **if** training **then**
        $(z, k, \mu) \leftarrow$ BANDKCORE($r, k_{\min}, k_{\max}$) {Alg. 2}
        $\pi \leftarrow \text{softmax}(r)$
        $m \leftarrow \text{stopgrad}(z - \mu) + \mu$ {Straight-through estimator over full vector, cf. Eq. (7)}
        $w^{\text{full}} \leftarrow m \odot \pi$
        $I \leftarrow$ TOPKINDICES($z, k_{\max}$)
        $w \leftarrow w^{\text{full}}|_I$
    **else**
        Find the MAP set of experts {Alg. 3}
    **end if**
    **return** $I, w$

---

---

**Algorithm 2** BANDKCORE$(r, k_{\min}, k_{\max})$

---

**input** Router logits $r \in \mathbb{R}^{T \times N}$, band $[k_{\min}, k_{\max}]$
**output** Sampled masks $S \in \{0, 1\}^{T \times N}$, sampled cardinalities $k \in \{k_{\min}, \ldots, k_{\max}\}^T$, marginals $\mu \in [0, 1]^{T \times N}$
$\quad \log p \leftarrow \log \sigma(r)$
$\quad \log q \leftarrow \log(1 - \exp(\log p))$
$\quad a \leftarrow \text{PRUPTOK}(\log p, \log q, E, k_{\max})$
$\quad k \leftarrow \text{SAMPLEBANDK}(a, k_{\min}, k_{\max})$
$\quad S \leftarrow \text{SAMPLEEXACTKDYNAMIC}(a, \log p, k)$
$\quad \mu \leftarrow \text{MARGINALSBAND}(r, k_{\min}, k_{\max})$
$\quad$**return** $S, k, \mu$

---

**function** PRUPTOK$(\log p, \log q, n, k_{\max})$
$\quad$ initialize $a[i, j] \leftarrow -\infty$ for all $i, j$
$\quad a[0, 0] \leftarrow 0$ {$\log A(0, 0) = \log 1 = 0; A(0, k) = 0$ for $k > 0$, cf. Prop. B.1}
$\quad$**for** $i = 1$ **to** $n$ **do**
$\quad\quad$**for** $j = 0$ **to** $k_{\max}$ **do**
$\quad\quad\quad a[i, j] = \text{logaddexp}\left(a[i-1, j-1] + \log p_i, \ a[i-1, j] + \log q_i\right)$
$\quad\quad$**end**
$\quad$**end**
$\quad$**return** $a$

**function** SAMPLEBANDK$(a, k_{\min}, k_{\max})$
$\quad \ell \leftarrow \left[a[n, k_{\min}], a[n, k_{\min} + 1], \ldots, a[n, k_{\max}]\right]$
$\quad \rho \leftarrow \text{softmax}(\ell)$
$\quad$ sample $j \sim \text{Categorical}(\rho)$
$\quad k \leftarrow j + k_{\min}$
$\quad$**return** $k$

**function** SAMPLEEXACTKDYNAMIC$(a, \log p, k)$
$\quad j \leftarrow k, \quad z \leftarrow [\,]$
$\quad$**for** $i = n$ **to** $1$ **do**
$\quad\quad \pi_i \leftarrow \left(a[i-1, j-1] + \log p_i\right) - a[i, j]$ {Log conditional: include expert $i$ given $j$ still needed}
$\quad\quad z_i \sim \text{Bernoulli}(\sigma(\pi_i))$
$\quad\quad$**if** $z_i = 1$ **then** $j \leftarrow j - 1$
$\quad\quad$ append $z_i$ to $z$
$\quad$**end**
$\quad$**return** $z$

**function** MARGINALSBAND$(r, k_{\min}, k_{\max})$
$\quad \log p \leftarrow \log \sigma(r)$
$\quad \log q \leftarrow \log(1 - \exp(\log p))$
$\quad a \leftarrow \text{PRUPTOK}(\log p, \log q, E, k_{\max})$
$\quad L \leftarrow \sum_t \text{logsumexp}\left(a[n, k_{\min}], \ldots, a[n, k_{\max}]\right)$
$\quad \mu \leftarrow \nabla_{\log p} L$
$\quad$**return** $\mu$

---

---

**Algorithm 3** Deterministic Top-$k$ Routing

---

**Input:** Logits $\theta \in \mathbb{R}^{B \times N}$, range limits $k_{\min}, k_{\max}$
**Output:** Indices $I \in \mathbb{Z}^{B \times k_{\max}}$, Weights $W \in \mathbb{R}^{B \times k_{\max}}$
$\pi \leftarrow \text{Softmax}(\theta, \text{axis}=1)$
$(I^{\text{all}}, s^{\text{sorted}}) \leftarrow \text{SortDescending}(\pi, \text{axis}=1)$ {$I^{\text{all}}$: sorted indices; $s^{\text{sorted}}$: corresponding sorted $\pi$ values}
$C \leftarrow \text{PrefixSum}(s^{\text{sorted}}, \text{axis}=1)$
$k^* \leftarrow \arg \max_{k \in \{k_{\min}, \dots, k_{\max}\}} C[:, k]$
$I \leftarrow I^{\text{all}}[:, 1 : k_{\max}]$
**for** $i = 1$ **to** $B$ **do**
    **for** $j = 1$ **to** $k_{\max}$ **do**
        **if** $j \leq k_i^*$ **then**
            $W_{i,j} \leftarrow \pi_{i, I_{i,j}}$
        **else**
            $W_{i,j} \leftarrow 0$
        **end if**
    **end for**
**end for**
**return** $I, W$

---

## H. ProbMoE Exact-$k$ Efficient SDD-Based Implementation

The chain-DP formulation in Algorithm 2 rebuilds its DP table from scratch at every forward pass. We additionally provide an optimized implementation that compiles the exactly-$k$ constraint once into a Sentential Decision Diagram (SDD) and reuses the compiled circuit thereafter. Both implementations are mathematically equivalent: they compute identical per-variable marginals and draw samples from the same constrained distribution.

We summarize the overall optimized ProbMoE fixed-$k$ framework in Algorithm 4, the core circuit-based sampling and marginal computation logic in Algorithm 5, and the deterministic top-$k$ routing procedure used for inference in Algorithm 7. The exactly-$k$-of-$n$ constraint is compiled once into a Sentential Decision Diagram (SDD) $\beta$ via Algorithm 6; the resulting circuit is reused for every forward pass at training time.

---

**Algorithm 4** $\text{PROBMOE-SDD}(H, k, \beta)$: Fixed-$k$ Circuit Routing for MoE

---

**input** Token states $H \in \mathbb{R}^{T \times d}$, router logits $r \in \mathbb{R}^{T \times N}$, top-$k$ value $k$, compiled SDD root $\beta$
**output** Selected experts $I$, routing weights $w$
  **if** training **then**
    $(z, \mu) \leftarrow \text{SDDCORE}(r, k, \beta)$ {Alg. 5}
    $\pi \leftarrow \text{softmax}(r)$
    $m \leftarrow \text{stopgrad}(z - \mu) + \mu$ {Straight-through estimator, cf. Eq. (7)}
    $w_{\text{full}} \leftarrow m \odot \pi$
    $I \leftarrow \text{TOPKINDICES}(z, k)$ {$z$ has exactly $k$ ones per row}
    $w \leftarrow w_{\text{full}}|_I$
  **else**
    $(I, w) \leftarrow$ deterministic top-$k$ routing {Alg. 7}
  **end if**
  **return** $I, w$

---

---

**Algorithm 5** SDDCORE($r, k, \beta$)

---

**input** Router logits $r \in \mathbb{R}^{T \times N}$, cardinality $k$, compiled SDD root $\beta$ encoding $\sum_i x_i = k$

**output** Sampled masks $z \in \{0, 1\}^{T \times N}$ with $\sum_i z_i = k$, marginals $\mu \in [0, 1]^{T \times N}$ where $\mu_{t,i} = \Pr[x_i = 1 \mid \sum_j x_j = k]$

  $\log p \leftarrow \log \sigma(r)$

  $\log q \leftarrow \log(1 - \exp(\log p))$ {Numerically-stable `log1mexp`; treated as stop-grad}

  $\log Z \leftarrow$ CIRCUITUPWARD($\log p, \log q, \beta$)

  $\mu \leftarrow \nabla_{\log p} \log Z$ {Marginal trick; via autograd or CIRCUITDOWNWARD}

  $z \leftarrow$ CIRCUITSAMPLE($\log p, \log q, \beta$)

  **return** $z, \mu$

---

**function** CIRCUITUPWARD($\log p, \log q, \beta$)

  **for** each literal node $\ell_i^+$ **do** $v[\ell_i^+] \leftarrow \log p_i$

  **for** each literal node $\ell_i^-$ **do** $v[\ell_i^-] \leftarrow \log q_i$

  **for** each level $L$ of $\beta$, leaves $\rightarrow$ root **do**

    **for** each decomposition node $n \in L$ with elements $\{(p_j, s_j)\}_j$ **do**

      $\theta[n, j] \leftarrow v[p_j] + v[s_j]$ {Element log-weights; $p_j, s_j$ are prime/sub children of $n$}

      $v[n] \leftarrow \text{logsumexp}_j \theta[n, j]$

      $\theta[n, \cdot] \leftarrow \theta[n, \cdot] - v[n]$ {Cache normalized log-conditionals}

    **end**

  **return** $v[\beta]$ {$\log Z$ = log-partition under constraint}

**function** CIRCUITDOWNWARD($\beta, \theta$) {Direct first-order marginals; alternative to autograd}

  $g[\beta] \leftarrow 0$;   $g[n] \leftarrow -\infty$ for all other nodes

  **for** each level $L$ of $\beta$, root $\rightarrow$ leaves **do**

    **for** each decomposition node $n \in L$ with elements $\{(p_j, s_j)\}_j$ **do**

      **for** each element index $j$ **do** {Distribute message from $n$ to its prime/sub children}

        $g[p_j] \leftarrow \text{logsumexp}\big(g[p_j], \ g[n] + \theta[n, j]\big)$

        $g[s_j] \leftarrow \text{logsumexp}\big(g[s_j], \ g[n] + \theta[n, j]\big)$

      **end**

    **end**

  $\mu_i \leftarrow \exp\big(g[\ell_i^+]\big)$ for each variable $i$

  **return** $\mu$

**function** CIRCUITSAMPLE($\log p, \log q, \beta$) {Exact ancestral sample from $\Pr[\cdot \mid \sum_i x_i = k]$}

  Run CIRCUITUPWARD to populate $\theta$

  $\text{active}[\beta] \leftarrow 1$;   $\text{active}[n] \leftarrow 0$ for all other nodes

  **for** each level $L$ of $\beta$, root $\rightarrow$ leaves **do**

    **for** each decomposition node $n \in L$ with $\text{active}[n] > 0$ **do**

      sample element index $j^* \sim \text{softmax}_j \theta[n, j]$

      $\text{active}[p_{j^*}] \mathrel{+}= 1$;   $\text{active}[s_{j^*}] \mathrel{+}= 1$ {Accumulate; subgraphs are shared}

    **end**

  $z_i \leftarrow \mathbb{1}[\text{active}[\ell_i^+] > 0]$ for each variable $i$

  **return** $z$

---

---

**Algorithm 6** COMPILEEXACTLYK$(n, k)$: One-time SDD construction

---

**input** Number of experts $n$ (power of two), cardinality $k$
**output** SDD root $\beta$ encoding $\sum_{i=1}^{n} x_i = k$
    create literal nodes $\ell_i^+, \ell_i^-$ for $i = 1, \ldots, n$
    $\mathrm{dp}_{\mathrm{prev}}[i, 0] \leftarrow \ell_i^-, \quad \mathrm{dp}_{\mathrm{prev}}[i, 1] \leftarrow \ell_i^+ \quad$ for $i = 1, \ldots, n$
    $m \leftarrow n$
    **while** $m > 1$ **do**
        $m \leftarrow m/2$
        **for** $i = 0$ **to** $m - 1$ **do**
            **for** $j = 0$ **to** $\min(k, n/m)$ **do**
                $\mathcal{E} \leftarrow \big\{ (\mathrm{dp}_{\mathrm{prev}}[2i, j'], \, \mathrm{dp}_{\mathrm{prev}}[2i + 1, j - j']) : 0 \leq j' \leq j \big\}$
                $\mathrm{dp}_{\mathrm{curr}}[i, j] \leftarrow$ LOOKUPORCREATE$(\mathcal{E})$ {Hash-cons to deduplicate shared subgraphs}
            **end for**
        **end for**
        $\mathrm{dp}_{\mathrm{prev}} \leftarrow \mathrm{dp}_{\mathrm{curr}}$
    **end while**
    **return** $\mathrm{dp}_{\mathrm{prev}}[0, k]$

---

**Algorithm 7** Deterministic Top-$k$ Routing (SDD inference)

---

    **Input:** Logits $\theta \in \mathbb{R}^{B \times N}$, fixed cardinality $k$
    **Output:** Indices $I \in \mathbb{Z}^{B \times k}$, Weights $W \in \mathbb{R}^{B \times k}$
    $\pi \leftarrow \mathrm{Softmax}(\theta, \ \mathrm{axis}=1)$
    $I \leftarrow \mathrm{TopKIndices}(\pi, \ k, \ \mathrm{axis}=1)$
    $W \leftarrow \mathrm{Gather}(\pi, \ I)$
    **return** $I, W$

---

