# OpenReview forum: "ProbMoE: Differentiable Probabilistic Routing for Mixture-of-Experts"
_ICML.cc/2026/Conference — ICML 2026 regular_

### Official Review · Reviewer_LwMj · 2026-03-01

**Soundness:** 3
**Presentation:** 3
**Significance:** 2
**Originality:** 2
**Overall Recommendation:** 4
**Confidence:** 3

**Summary:**

This paper proposes a technique to address the issue of non-differentiability of the router parameters induced by the top-k operator which enables sparse training/inference. The key idea is to reformulate the expert selection procedure as a stochastic latent variable problem where the network produces a distribution of expert subsets (of the same cardinality i.e k) for each token followed by inference over these subsets which is really just selecting one of these subsets for the actual routing procedure. The authors first go on to develop an exact-k subset selection scheme but then move towards a dynamic-k setup where each token may be assigned a varying number of experts. The authors present a number of experiments where they demonstrate the effectiveness of their technique on fine-tuning datasets and Qwen and OlMoE and compare their approach to several baselines from prior work.

**Compliance With Llm Reviewing Policy:**

Affirmed.

**Final Justification:**

The authors addressed the gaps in their evaluation by adding MMLU and coding tasks, which shows the performance gains are consistent across different benchmarks. Moving away from hacky STE methods toward a more principled probabilistic approach for routing is a solid step forward, and the results now sufficiently back up the framework. Moving to Weak Accept.

**Key Questions For Authors:**

How does the runtime / overhead compare to w.r.t a baseline and standard top-k?

I’m not sure where the results for the dynamic-k subset selection technique are and whether they are better/worse. Also i’d like to see how much improvement in wall clock time does the dynamic k scheme provide as i see it leads to fewer experts being used overall

**Limitations:**

yes

**Strengths And Weaknesses:**

Strengths
The paper has been written very well and is easy to follow. Readers will appreciate the background on MoEs as well as what exactly makes conventional top-k routing non-differentiable. Their framework is principled and their analysis of it seems correct. They experiment with multiple models and include compare their method to several other works all of which deal with the issue of nondifferentiability of conventional MoEs and show that their method improves upon the baseline for GSM, Law and a translation task for at least one of their models.



Weaknesses
I can see two main weaknesses with this paper. One is the evaluation has been conducted on a very few benchmarks and i’d really like to see how their technique improves on some of the more comprehensive benchmark scores like HumanEval, MMLU and some math / reasoning benchmarks. Secondly, even for the datasets they do show results on, the improvements are quite marginal and not consistent. For instance, DenseMixer is already very close to their method and beats their methodology for one of the datasets for both models. This makes me wonder how much improvement does their framework really bring upon compared to basic STE based techniques.

---

> ### Author Rebuttal · Authors · 2026-03-31
>
> We appreciate the reviewer's valuable feedback. Below we provide our responses including additional experiment results and clarifications.
>
> *[Q1 Runtime/overhead compare]*
>
> For a detailed wall-clock/memory table and analysis, please refer to our response to Reviewer cGjY[w1/Q2]. To summarize briefly: SIMoE incurs zero additional cost at inference compared to conventional methods, and during training, it remains faster and more memory efficient than DenseMixer per epoch wise. Given the consistent routing quality improvements demonstrated in our results, we believe this training overhead is well justified.
>
> *[W1& W2 More benchmark scores & marginal Improvements]*
>
> As suggested by the reviewer, we expand our evaluations to include additional benchmarks, including CodeAlpaca [4] and MBPP [2] for code generation, MMLU [3] for general knowledge retention, and Summary [1] for natural language understanding. As shown in Table 3 in our response to Reviewer U4vc section [W1/Q1], SIMoE consistently outperforms other methods across all benchmarks.
>
> *Table 4: Percent change in performance when using SIMoE compared to other models.*
>
> |Dataset|SIMoE vs Conv.|SIMoE vs D.M.|
> |:---|:---|:---|
> |OLMoE Backbone (Top-k = 8)|||
> |GSM|+9.25|+2.43|
> |LAW|+29.75| +25.98|
> |Translation|+17.63|-2.83|
> |Qwen Backbone (Top-k = 4)|||
> |GSM|-0.24|-3.39|
> |LAW|+17.61|+1.68|
> |Translation|+18.81|+15.72|
>
> We would like to highlight that SIMoE achieves consistent improvements across settings: as shown in Table 4 above, and Table 3 in our response to Reviewer U4vc section [W1/Q1], SIMoE outperforms both Conventional and DenseMixer on 6 out of 7 benchmarks, with substantial gains on Translation, Summary, and Law datasets. SIMoE is also the only method to improve MMLU-stem over the conventional baseline (53.82 vs. 53.16), whereas DenseMixer slightly degrades it (52.87).
>
> More importantly, SIMoE’s core contribution extends beyond improving fixed-K routing performance. By formulating expert selection as a principled probabilistic distribution over discrete subsets, rather than relying on the deterministic hard thresholding of STE based methods, SIMoE enables dynamic routing by nature, allowing the model to adaptively select a different number of experts per token and per layer, while the current methods like DenseMixer or any STE-based method are strictly limited into a fixed top-K selection.
>
> *[Q2 Results for the dynamic-k]*
>
> We refer the reviewer to Table 2 in the submission, which reports the performance gains of SIMoE-Dynamic across benchmarks, with SIMoE-Exact serving as the baseline. SIMoE-Dynamic activates a smaller fraction of experts per token compared to the full top-K set, yet achieves performance that is either competitive with or in several cases surpasses SIMoE-Exact. This demonstrates that adaptively selecting fewer experts per layer and per token can maintain, or even improve, model quality while activating less experts.
>
> *[Q2 Improvement in wall clock time in dynamic k]*
>
> SIMoE-Dynamic does indeed activate fewer experts overall, as different layers and tokens are assigned different numbers of active experts based on the learned routing distribution. In principle, this variable activation pattern should translate to wall-clock speedups at inference, since fewer experts means fewer FLOPs. However, realizing these gains in practice requires nontrivial system implementations. Current MoE serving frameworks, including HF Transformers and vLLM, assume a fixed top-K per token. While all expert weights are loaded into GPU memory regardless of how many are activated, the compute kernels, such as FusedMoE and grouped GEMM [5], allocate activation buffers, sort tokens, and batch matrix multiplications assuming every token activates exactly K experts. Adapting these kernels to support variable-K routing is nontrivial; for instance, an attempt to implement dynamic expert allocation in vLLM resulted in numerical instability. We view this as an open and promising direction: SIMoE-Dynamic provides the algorithmic foundation for adaptive expert activation, and we believe that pairing it with optimized serving infrastructure that supports variable-K routing could unlock meaningful wall-clock improvements at inference. We leave this system level exploration as future work.
>
> We hope you find our response with additional experiments and further clarification to be helpful, and consider turning the verdict into an accept.
>
> [1] Let the expert stick to his last: Expert-specialized fine-tuning for sparse architectural large language models. EMNLP 2024. Wang et.al.
>
> [2] Program synthesis with large language models. 2021. Austin et.al.
>
> [3] Measuring massive multitask language understanding. LCLR 2021. Hendrycks et.al.
>
> [4] https://github.com/sahil280114/codealpaca
>
> [5] https://github.com/vllm-project/vllm/issues/17150

---

> > ### Author Rebuttal · Reviewer_LwMj · 2026-04-03
> >
> > Thank you for the results. I have adjusted my scores accordingly.

---

> > > ### Author Response · Authors · 2026-04-06
> > >
> > > Thank you for your positive feedback and increasing the score!

---

### Official Review · Reviewer_U4vc · 2026-03-08

**Soundness:** 2
**Presentation:** 3
**Significance:** 2
**Originality:** 2
**Overall Recommendation:** 3
**Confidence:** 4

**Summary:**

This paper proposes SIMoE, a novel probabilistic routing framework for MoE models. It discusses the problem arising from the non-differentiable deterministic top-k routing operator and the reliance on surrogate gradients. To address this, the work investigates the concept of modeling expert selection as a stochastic latent variable. By formulating routing as probabilistic inference over discrete expert subsets, the authors leverage exact inclusion marginals to propagate gradients.
Empirical evaluations on OLMoE and Qwen MoE models across GSM8K, Law, and Translation tasks demonstrate that SIMoE improves expert utilization, routing diversity, and achieves competitive or superior performance.

**Compliance With Llm Reviewing Policy:**

Affirmed.

**Final Justification:**

This paper addresses a problem in MoE fine-tuning by proposing a probabilistic routing formulation over discrete expert subsets. The authors’ rebuttal has helped clarify the methodological novelty of the work relative to prior adaptive routing approaches. However, the comparison with related methods remains largely conceptual, and the paper still lacks direct empirical evidence to substantiate the claimed advantages of the proposed method, especially given that prior methods can achieve similar Dynamic-k behavior in practice.
As a result, my overall assessment is weak reject.

**Key Questions For Authors:**

1. Could you provide additional evaluation results across a more diverse and comprehensive set of downstream benchmarks to better assess the model's generalization capabilities?
2. Could the fine-tuning experimental setup be expanded? Specifically, evaluating the model after fine-tuning on commonly used SFT datasets across multiple downstream tasks would be highly informative. Furthermore, the manuscript notes that SIMoE occasionally requires additional training steps to reach stable convergence. This delayed convergence could potentially compromise the general-purpose capabilities of foundation models in real-world LLM production.
3. Could you demonstrate the efficacy of the proposed method in a from-scratch pre-training setting, similar to the experiments utilized in DefaultMoE paper?

**Limitations:**

yes

**Strengths And Weaknesses:**

Strengths:
1. The manuscript is well-structured, clearly written, and easy to follow, making the theoretical formulations and algorithmic designs accessible to the reader.
2. The paper provides a thorough discussion of related works, adequately contextualizing the proposed method within the current landscape of MoE routing optimization and establishing a clear baseline for comparison.

Weaknesses:
1. The current experimental design and the selection of benchmarks are somewhat restricted in scope. The evaluation is not broad enough to fully demonstrate the method's generalizability.
2. Although the proposed probabilistic approach is theoretically appealing, its fundamental objectives closely align with existing strategies. Moreover, the empirical results do not demonstrate a statistically significant or compelling performance improvement over established baselines.

---

> ### Author Rebuttal · Authors · 2026-03-31
>
> Thank you for your time and feedback. Below we provide our responses including additional experiment results and clarifications.
>
> *[W1/Q1 Extra Eval Results]*
>
> *Table 3: Performance comparison of models using the Qwen-2.7B-14B.* Baseline results taken from: https://github.com/yaof20/DenseMixer/issues/10
>
> |Model|Summary|MBPP|MMLU-overall|MMLU-stem|
> |:---|:---|:---|:---|:---|
> |Conventional|39.00|32.80|61.03|53.16|
> |DenseMixer|41.00|34.00|61.03|52.87|
> |SIMoE|**44.40**|**35.00**|**61.05**|**53.82**|
>
> To assess the generalizability of SIMoE, we expand our evaluation to include two highly contrasting domains: natural language understanding and structured programming. Specifically, we add the Summary (19.6K)[7] dataset to test semantic text compression, alongside CodeAlpaca (22K)[10] to test formal logic and code generation. Testing across distinct linguistic and structural boundaries provides a much stronger measure of the router's adaptability. For the coding domain, we finetune on CodeAlpaca and evaluate on MBPP[8] for code generation and MMLU[9] for general knowledge retention. We also report MMLU-STEM as a relevant subset given the code oriented nature of CodeAlpaca. As shown in the above table, SIMoE consistently outperforms both baselines across the new benchmarks.
>
> *[Q2: Extra Downstream Tasks&Delayed Convergence]*
>
> The additional MMLU experiments, as suggested by the reviewer, confirm that improved routing preserves general knowledge retention even after domain-specific finetuning. Additionally, SIMoE also improves MMLU-STEM over the conventional baseline, whereas DenseMixer slightly degrades it.
>
> For a detailed wall-clock analysis, please refer to our response to Reviewer DpaL[W1]. In short, SIMoE adds no inference cost and is faster and more memory efficient than DenseMixer per epoch and the training overhead is well justified by the consistent routing quality improvements.
>
> *[W2: Fundamental objectives & statistically significant results]*
>
> To our knowledge, SIMoE is the first to formulate MoE routing as probabilistic inference over discrete expert subsets under cardinality constraints. This formulation is what fundamentally distinguishes SIMoE from prior work: by defining a proper distribution over k-subsets, SIMoE naturally extends to Dynamic-k routing, where the number of active experts is itself a latent variable inferred per token per layer. This flexibility is structurally unavailable to STE-based methods, which are locked into fixed top-K selection. We would also like to highlight that SIMoE achieves noticeable improvements in many settings, as shown in Table 4 in our response to Reviewer LwMj [W1&W2].
>
> *[Q3: The efficacy of pre-training setting]*
>
> The scope of our work is the fine-tuning setting, where critical challenges arise: the non-differentiable Top-K routing blocks gradient flow to inactive experts[6], and adapting to new domains frequently triggers expert collapse and routing concentration[3]. SIMoE joins a growing line of work dedicated to improving MoE routing in the post-training regime[1,2,4,5], and we view pre-training and post-training routing enhancements as orthogonal, each addressing the routing problem under different constraints. Our analysis in the submission (Figures 3&4) highlights this vulnerability, showing that Conventional and DenseMixer routing collapse into highly concentrated within-layer distributions during adaptation. In contrast, SIMoE maintains more balanced expert utilization, demonstrating that principled routing interventions are essential within the fine-tuning phase.
>
> We appreciate the reviewer’s suggestion to evaluate SIMoE in a from-scratch pre-training setting. We consider this an important next step, as SIMoE's probability-based optimization provides dense gradient signals despite sparse expert selection, giving it strong theoretical potential to stabilize early pre-training dynamics.
>
> We hope you find our response with additional experiments and further clarification to be helpful, and consider turning the verdict into an accept.
>
> [1]Auxiliary-loss-free load balancing strategy for mixture-of-experts. 2024. Wang et.al.
>
> [2]CoMoE: Contrastive Representation for Mixture-of-Experts inParameter-Efficient Fine-tuning. ACL 2025. Feng et.al.
>
> [3]EPnG: Adaptive Expert Prune-and-Grow for Parameter-Efficient MoE Fine-tuning. Lee et.al.
>
> [4]On the Representation Collapse of Sparse Mixture of Experts. NeurIPS 2022 Chi et.al.
>
> [5]Parameter-Efficient Routed Fine-Tuning: Mixture-of-Experts Demands Mixture of Adaptation Modules. EACL 2026. Liu et.al.
>
> [6]ReMoE: Fully Differentiable Mixture-of-Experts with ReLU Routing. ICLR 2025. Wang et.al.
>
> [7]Let the expert stick to his last: Expert-specialized fine-tuning for sparse architectural large language models. EMNLP. Wang et.al.
>
> [8]Program synthesis with large language models. Austin et.al.
>
> [9]Measuring massive multitask language understanding. LCLR. Hendrycks et.al.
>
> [10]https://github.com/sahil280114/codealpaca

---

> > ### Author Rebuttal · Reviewer_U4vc · 2026-04-03
> >
> > Thank you for the clarification and additional discussion. Your response partially addresses my concerns.
> >
> > However, I am still not fully convinced by the argument regarding fundamental objectives.
> > I agree that SIMoE appears to be the first work to cast MoE routing as probabilistic inference over discrete expert subsets under cardinality constraints. However, the ability to perform Dynamic-k routing is not unique to this formulation. A number of prior works have already explored routing schemes with an adaptive number of active experts, for example:
> > https://arxiv.org/pdf/2403.07652,
> > https://openreview.net/pdf?id=9CqkpQExe2,
> > https://arxiv.org/abs/2410.07348,
> > https://arxiv.org/pdf/2512.13996v1.

---

> > > ### Author Response · Authors · 2026-04-06
> > >
> > > Thank you for acknowledging our work to be the first work to cast MoE routing into a probabilistic framework and the pointers. By formulating MoE routing as probabilistic inference over discrete expert subsets under cardinality constraints, we define a proper distribution over the combinatorial space of expert selections, and further demonstrate that it can be trained end-to-end with our proposed gradient estimators.
> > >
> > > This probabilistic framework naturally unifies Exact-k and Dynamic-k routing in a principled way. In contrast, prior adaptive expert allocation methods are highly heuristic, as discussed in the "Adaptive Expert Allocation" paragraph in the related work section in our submission. To further clarify the distinction, we provide comparison between our work and the existing work you mentioned below:
> > >
> > > * Huang et al. [1], which has been cited in our paper, employ a confidence-based heuristic: experts are greedily accumulated until their cumulative routing probability exceeds a predefined threshold $p$. Thus, the number of active experts is sensitive to the choice of the hyperparameter $p$. Moreover, their method requires *pre-training* from scratch with the dynamic routing mechanism built in. *SIMoE* instead treats the number of active experts as *a latent variable* inferred through *probabilistic inference* over expert subsets, associated with the probability of each expert subset and jointly with the subset selection itself, and can be applied to existing pre-trained MoE models during *fine-tuning*.
> > >
> > > * Ada-K [2] trains a *separate* RL-based allocator module via Proximal Policy Optimization to predict the number of experts per token. The allocator is a post-hoc module trained on top of *frozen* pre-trained MoE models. Here, the number of experts is the output of a discrete reinforcement learning policy with an external reward signal, decoupled from the routing objective itself.  In *SIMoE*, the decisions of *how many* and *which* experts to activate are coupled through *a single probabilistic objective* and trained end to end, rather than requiring a separate module with its own optimization procedure.
> > >
> > > * MoE++ [3] augments the *expert pool* with zero-computation pseudo-experts alongside standard FFN experts, and applies fixed top-K routing over this combined pool. While the effective number of FFN experts engaged varies per token (since some selected slots may be zero-computation), the router itself *still performs a fixed top-K* selection over the augmented set. Dynamic compute is thus an architectural side-effect of pool composition. In *SIMoE*, variable expert counts arise not from augmenting the expert pool, but from the *routing mechanism itself*, which infers a posterior over the combinatorial space of expert subsets with varying cardinalities. This also means SIMoE can be applied to any existing pre-trained MoE model during *fine-tuning*, whereas MoE++ requires *pre-training* from scratch with the augmented expert pool.
> > >
> > > * DTop-p [4] uses a Proportional-Integral controller from classical control theory to dynamically adjust a global probability threshold during training. The number of active experts is regulated by an external feedback control loop, not by posterior inference. Moreover, DTop-p is designed for *pre-training* from scratch. In *SIMoE*, the number of active experts is instead determined by a learned posterior over subset cardinalities, the selection within the range is *governed by probabilistic inference* rather than *an external controller*, and can be applied to pre-trained MoE models during *fine-tuning*.
> > >
> > > In summary, while all methods discussed above achieve adaptive expert counts, none defines a proper probability distribution over expert subsets or treats the number of active experts in a principled way. Moreover, all four methods require either pre-training from scratch or a separate post-hoc training stage, whereas SIMoE can be directly applied to existing pre-trained MoE models during fine-tuning. Thus the distinction is structural, not terminological. We will include the above discussion in the final version. In light of our response, we respectfully ask that the paper be evaluated with full consideration of the scope and depth of its contributions as a whole and that you will consider raising your score.
> > >
> > > [1] Harder Tasks Need More Experts: Dynamic Routing in MoE Models. ACL2024. Huang et.al.
> > >
> > > [2] ADA-K ROUTING: BOOSTING THE EFFICIENCY OF MOE-BASED LLMS. ICLR 2025. Yue et.al.
> > >
> > > [3] MoE++: Accelerating Mixture-of-Experts Methods with Zero-Computation Experts. ICLR 2025. Jin et.al
> > >
> > > [4] Sparsity-Controllable Dynamic Top-p MoE for Large Foundation Model Pre-training. 2025. Jin et.al.

---

### Official Review · Reviewer_cGjY · 2026-03-08

**Soundness:** 4
**Presentation:** 4
**Significance:** 3
**Originality:** 3
**Overall Recommendation:** 6
**Confidence:** 3

**Summary:**

In this paper, this authors propose a sparse mixture-of-experts (MoE) routing framework called SIMOE that allows one to include the probability of selecting a certain subset of experts into the backpropagated gradients. Conventional approaches typically only train the router through its effect on the expert aggregation weights and ignore the part about the router's impact on the selection itself. The authors do this by leveraging an insight from SIMPLE (Ahmed et al. 2023) that derived explicit, differentiable forms for the probability of the selection of certain subsets. They explicitly show the realization of this framework for k-expert routing as well as dynamic routing within a band. The tradeoff of this framework is that one is nolonger choosing the top experts during training, but instead sampling the choice of experts. The sampling mechanism itself (described by the authors in their appendix) incurs an extra O(experts * selection_size) cost from the need to solve a dynamic program. The authors shows the advantage of their method when finetuning across two MoE architectures (Qwen and OLMoE) and three tasks (GSM, Law, Translation).

**Compliance With Llm Reviewing Policy:**

Affirmed.

**Final Justification:**

As stated before: This paper presents a clear idea of leveraging SIMPLE for sparse MoE routing. The mathematical derivations are clearly presented and the algorithms are clear to understand. The experiments are scientific and honest: Demonstrating a the general advantage of SIMOE, but also being transparent about the fact that it is not always be best (see Table 1). The authors also include an insightful ablation study showing the relative contributions of different components of their proposal (Figure 2).

In their rebuttal, the authors also satisfactorily addressed my concerns, for example about post- vs pre-training aspects, and even provided additional experiments. Given their effort, I have raised their score to a 6.

**Key Questions For Authors:**

(1) Might the authors clarify the pre- vs post-training aspects of the experiments? In particular, how much the finetuning experiments are indicative of potential advantages in pre-training.

(2) Might you clarify what how much the overhead of the DP affect the wallclock time?

**Limitations:**

Somewhat. The authors are clear about most aspects of their experiments. But, there seems to be tradeoffs that should be more explicitly discussed and also it seems most experiments seem to be in the finetuning regime which is less sensitive to MoE imbalance than the from-scratch training regime.

**Strengths And Weaknesses:**

Strength:

This paper presents a clear idea of leveraging SIMPLE for sparse MoE routing. The mathematical derivations are clearly presented and the algorithms are clear to understand. The experiments are scientific and honest: Demonstrating a the general advantage of SIMOE, but also being transparent about the fact that it is not always be best (see Table 1). The authors also include an insightful ablation study showing the relative contributions of different components of their proposal (Figure 2).

Weaknesses:

(1) It is unclear whether the dynamic program needed to do the sampling (Appendix D) incurs extra costs in terms of wall clock times to make the implementation of SIMoE worth it.

(2) From Table 4 (which says finetuning-only) and the fact that the baseline DenseMixer is a post-training method, my understanding is that all experiments in this paper are FINETUNING experiments on existing MoE checkpoints (rather than pre-training from a random initialization). If my understanding is correct, then the MoEs would start out "already roughly balanced" from the pre-training procedure and the contribution of SIMoE is restricted to improving performance in the finetuning. Additionally, many issues with MoE balancing occur during the pre-training  stage due to high variance in the routing logits and high instability in expert performance at initialization. Thus, the (implied) restriction of the experiments to only the finetuning regime may not be representative of SIMoE's relative performance overall across in the unexplored pre-training setting.

---

> ### Author Rebuttal · Authors · 2026-03-31
>
> Thank you for your positive feedback! Below we provide our responses including additional experiment results and clarifications.
>
> *[W1/Q2: Wall-clock times]*
>
> The dynamic program in Appendix D is used exclusively during training and incurs zero cost at inference, where both SIMoE-Exact and SIMoE-Dynamic simply select the Maximum A Posteriori (MAP) set. Thus, SIMoE-Exact is identical to conventional MoE routing during inference.
>
> During training, as shown in Table 1 below, the dynamic program does introduce additional per epoch overhead compared to the conventional baseline, but SIMoE’s per epoch wall-clock time and peak GPU memory remain comparable to the conventional baseline and consistently lower than DenseMixer across datasets as Densemixer requires a dense forward pass through all experts for every token to compute its straight-through gradient, dramatically increasing forward-pass FLOPs. In contrast, SIMoE’s routing optimization operates over $N$ scalar routing logits with $O(N \cdot k_{max})$ complexity, which is substantially cheaper than running full FFN computations for all inactive experts. Given the per epoch cost and the consistent improvements in routing quality demonstrated in our results, the overhead of the dynamic programming procedure is well justified.
>
> *Table 1: Training time per epoch and peak GPU memory per GPU (x4) OLMoE-1B-7B.*
>
> |Dataset|Conventional Time|Conventional Mem.|DenseMixer Time|DenseMixer Mem.|SIMoE Time|SIMoE Mem.|
> |:---|:---|:---|:---|:---|:---|:---|
> |GSM (7.4K)|35.58 min|46.0 GB|60.2 min|69.2 GB|49.8 min|46.4 GB|
> |Law (1K)|5.1 min|65.0 GB|16.5 min|74.8 GB|9.25 min|68.0 GB|
> |Translation (11K)|110.0 min|43.4 GB|111.8 min|64.4 GB|120.7 min|50.0 GB|
>
> *[W2/Q1: Pre- vs post-training aspects of the experiments]*
>
> All experiments in this paper are fine-tuning experiments on existing pre-trained MoE checkpoints.
>
> While pre-trained MoE checkpoints often begin with roughly balanced routing, critical challenges emerge during post-training. First, the non-differentiable Top-K routing mechanism blocks gradient flow to inactive experts, preventing the router from exploring alternative configurations and leaving most expert parameters without learning signal during fine-tuning [3,7]. Second, adapting to new domains or shifted input distributions frequently triggers expert collapse and routing concentration, where the router converges to overutilize a small subset of experts [4]. Even small distribution shifts during post-training can cascade into large, destabilizing changes in routing decisions.
>
> We specifically aim to solve those routing degradation. Because this adaptation phase requires dedicated interventions, the scope of our work is to the post-training regime using existing pre-trained MoE checkpoints. SIMoE addresses this by formulating expert selection as a differentiable optimization problem, enabling principled updates to routing decisions without relying on biased gradient estimators. As shown in Figure 3&4 in our submission, standard fine-tuning with Conventional or DenseMixer routing leads to highly concentrated within-layer distributions dominated by fewer experts. In contrast, our proposed SIMoE maintains more broad, balanced expert utilization, situating our contributions alongside a growing body of literature dedicated to correcting MoE routing post-training [1, 2, 5, 6].
>
> Meanwhile, we fully agree with the reviewer that there is compelling potential for SIMoE in the pre-training setting. SIMoE's marginal-based optimization, which provides dense gradient signals even under sparse expert selection, could be particularly beneficial during pre-training, where routing logits are highly unstable and expert specialization has not yet emerged. Ultimately, our insights and methodology open new possibilities for pre-training MoEs. However, rigorously validating this would require training large-scale MoE models from random initialization, which is beyond our current computational resources. We therefore leave this as an important avenue for future work, and believe the strong fine-tuning results presented here provide solid motivation for that exploration.
>
> We hope you find our response with additional experiments and clarification to be helpful and have addressed your concerns.
>
> [1]Auxiliary-loss-free load balancing strategy for mixture-of-experts. 2024. Wang et.al.
>
> [2]CoMoE: Contrastive Representation for Mixture-of-Experts inParameter-Efficient Fine-tuning. ACL 2025. Feng et.al.
>
> [3]DenseMixer: Improving MoE Post-Training with Precise Router Gradien. 2026. Yao et.al.
>
> [4]EPnG: Adaptive Expert Prune-and-Grow for Parameter-Efficient MoE Fine-tuning. Lee et.al.
>
> [5]On the Representation Collapse of Sparse Mixture of Experts. NeurIPS 2022 Chi et.al.
>
> [6]Parameter-Efficient Routed Fine-Tuning: Mixture-of-Experts Demands Mixture of Adaptation Modules. EACL 2026. Liu et.al.
>
> [7]ReMoE: Fully Differentiable Mixture-of-Experts with ReLU Routing. ICLR 2025. Wang et.al.

---

> > ### Author Rebuttal · Reviewer_cGjY · 2026-04-03
> >
> > The authors addressed my concerns and went above-and-beyond in providing additional measurements and citations. I appreciate this.

---

> > > ### Author Response · Authors · 2026-04-06
> > >
> > > Thank you for your positive feedback and increasing the score!

---

### Official Review · Reviewer_DpaL · 2026-03-18

**Soundness:** 3
**Presentation:** 3
**Significance:** 2
**Originality:** 3
**Overall Recommendation:** 4
**Confidence:** 3

**Summary:**

This paper proposes SIMoE, which considers MoE routing as probabilistic inference over discrete expert subsets. It builds on SIMPLE to define a Bernoulli subset distribution conditioned on cardinality constraints. It yields tractable normalization and closed-form inclusion marginals. Gradients flow through these marginals, giving the router dense signals while keeping forward execution sparse. In experiments, the authors evaluated it on OLMoE-1B-7B and Qwen1.5-MoE-A2.7B across GSM8K, Law, and Translation, with analysis showing broader expert utilization.

**Compliance With Llm Reviewing Policy:**

Affirmed.

**Key Questions For Authors:**

1. Are auxiliary balancing losses active in your experiments?
2. Have you measured variance of the STE-based routing weight estimator compared to DenseMixer?

**Limitations:**

computational overhead

**Strengths And Weaknesses:**

**Strengths**

- The paper clearly explains why top-k breaks gradient flow, then derives an alternative where training and inference operate on the same discrete distribution.
- Building on SIMPLE keeps the contribution focused.
- The experiments are well-designed, and the results are well-explained.

**Weaknesses**
- The paper acknowledges needing extra epochs for convergence but reports zero wall-clock or FLOPs numbers. Practitioners can't evaluate feasibility without this.
- Standard MoE training uses auxiliary balancing losses. The paper never discusses whether these are active, how SIMoE interacts with them.

---

> ### Author Rebuttal · Authors · 2026-03-31
>
> Thank you for your positive review! Below we provide our responses including additional experiment results and clarifications.
>
> *[W1: Wall-clock/Computational overhead]*
>
> *Inference Overhead:*
>
> SIMoE introduces no additional computational cost at inference time. Both SIMoE Exact and dynamic settings select the MAP (Maximum A Posteriori) set at inference. Thus, for SIMoE-Exact, the MAP set reduces to selecting the top-K experts, which is identical to conventional MoE routing. SIMoE-Dynamic, on the other hand, selects a MAP set that activates fewer experts than Top-$K$, theoretically making it more efficient at inference.
>
> *Training Overhead:*
>
> *Table 1: Training time per epoch and peak GPU memory per GPU (x4) OLMoE-1B-7B.*
>
> |Dataset|Conventional Time|Conventional Mem.|DenseMixer Time|DenseMixer Mem.|SIMoE Time|SIMoE Mem.|
> |:---|:---|:---|:---|:---|:---|:---|
> |**GSM (7.4K)**|35.58 min|46.0 GB|60.2 min|69.2 GB|49.8 min|46.4 GB|
> |**Law (1K)**|5.1 min|65.0 GB|16.5 min|74.8 GB|9.25 min|68.0 GB|
> |**Translation (11K)**|110.0 min|43.4 GB|111.8 min|64.4 GB|120.7 min|50.0 GB|
>
> As shown in Table 1, SIMoE’s per epoch wall-clock time and peak GPU memory remain comparable to the conventional baseline and consistently lower than DenseMixer across datasets as Densemixer requires a dense forward pass through all experts for every token to compute its straight-through gradient, dramatically increasing forward-pass FLOPs. In contrast, SIMoE’s routing optimization operates over $N$ scalar routing logits with $O(N \cdot k_{max})$ complexity, which is substantially cheaper than running full FFN computations for all inactive experts. While on some datasets, SIMoE requires additional epochs to converge, the overall training overhead remains manageable in practice.
>
> *[W2/Q1: Auxiliary balancing loss]*
>
> No auxiliary balancing losses are used in any of the experiments, as we follow the current convention in MoE post-training where no auxiliary losses are applied [1,2,3,4,5]. Investigating the role of auxiliary balancing losses in MoE post-training can be an important direction for future work, albeit orthogonal to the focus of this work.
>
> Additionally, SIMoE achieves effective expert utilization without requiring explicit load balancing losses, through a fundamentally different mechanism. As shown in Figure 3&4 in our submission, SIMoE maintains a broader routing distribution than both Conventional and DenseMixer routing, with higher normalized entropy across layers, indicating that expert traffic is distributed more evenly. We attribute this to SIMoE's marginal based optimization. This broader expert participation naturally mitigates expert collapse and reduces routing concentration, without requiring any explicit regularization.
>
> *[Q2: Gradient variance comparison]*
>
> *Table 2: Comparison of aggregated bias, variance, and gradient error between SIMoE and DenseMixer on a synthetic MoE task.*
>
> |Model|Aggregated Bias|Aggregated Variance|Aggregated Error|
> |:---|:---|:---|:---|
> |**SIMoE**|**0.0702 ± 0.0426**|**0.0076 ± 0.0110**|**0.0967 ± 0.0659**|
> |DenseMixer|0.1052 ± 0.0839|0.0201 ± 0.0427|0.1349 ± 0.0995|
>
>
> Calculating exact ground-truth gradients on real data is intractable. In order to accurately measure an estimator's bias and variance for a comprehensive comparison, we consider a controlled setting where the true gradient is known. We carry out a synthetic experiment with a top-$K$ distribution over $n$=10 experts and $k$=5, yielding $C(10,5)=252$ possible subsets, making the exact gradient tractable in closed form. The setup simulates $T=10$ tokens with per token router logits to mimic real MoE behavior, and compare DenseMixer against SIMoE. We measure bias, variance, and average error via cosine distance , a widely adopted metric for assessing gradient variance [6], against the exact gradient over 10,000 samples across 10 random seeds. As shown in Table 2, SIMoE consistently outperforms DenseMixer across all three metrics, achieving lower variance (0.008 vs. 0.020), lower bias (0.070 vs. 0.105), and lower overall gradient error (0.097 vs. 0.135), confirming that SIMoE's marginal-based estimator provides a more accurate and stable approximation of the true gradient.
>
> We hope you find our response with additional experiments and further clarification to be helpful, and consider increasing your score in the final evaluation.
>
>
> [1] Auxiliary-Loss-Free Load Balancing Strategy for Mixture-of-Experts. 2024. Wang et.al.
>
> [2] CoMoE: Contrastive Representation for Mixture-of-Experts inParameter-Efficient Fine-tuning. EMNLP 2025. Feng et.al.
>
> [3] DeepSeek-V2: A Strong, Economical, and Efficient Mixture-of-Experts Language Model. CoRR 2024. DeepSeek-AI, et.al.
>
> [4] DenseMixer: Improving MoE Post-Training with Precise Router Gradien. 2026. Yao et.al.
>
> [5] MoECondenser: Finetuning MoE LLMs with Condenser Experts. 2026 He et.al.
>
> [6] SIMPLE: A Gradient Estimator for k-Subset Sampling. ICLR 2023. Ahmed et.al.

---

> > ### Author Rebuttal · Reviewer_DpaL · 2026-04-04
> >
> > My concerns have been addressed.

---

> > > ### Author Response · Authors · 2026-04-06
> > >
> > > We are glad that our responses have addressed your concerns!

---

### Decision · Program_Chairs · 2026-04-30

**Decision:**

Accept (regular)

**Comment:**

This paper proposes a sparse mixture-of-experts (MoE) routing framework, SIMoE, that incorporates the probability of selecting a given subset of experts into the backpropagated gradients. The key technical ingredient is the use of results from SIMPLE, which provide explicit and differentiable expressions for subset-selection probabilities under cardinality constraints. The method is evaluated when fine-tuning two MoE architectures, Qwen and OLMoE, on three tasks: GSM, Law, and Translation.

The reviewers agree that the underlying idea is relevant and interesting. The paper addresses a real issue in MoE training, namely the mismatch between discrete routing decisions at inference time and the surrogate gradients typically used during training. At the same time, the main idea is relatively simple and natural once stated, and the overall conceptual novelty appears somewhat limited. The theoretical contribution is also not particularly strong, as the paper mainly builds on an existing probabilistic insight rather than developing a substantially new theoretical framework.

The main concerns raised by the reviewers relate to the experimental validation. In its original form, the evaluation was considered too limited, both in the number of tasks and in its exclusive focus on fine-tuning rather than pre-training. The authors have partly addressed this point by adding results on MMLU and coding tasks, but the scope of the paper remains on fine-tuning only. A second concern is that the empirical gains over existing methods remain relatively modest. While this does not invalidate the method, it limits the overall impact of the paper. Finally, the reviewers also questioned the positioning with respect to related work and the benchmark choices, especially for the Dynamic-k routing setting, where comparisons could be more convincing.

Overall, the paper presents a sound and relevant idea, and the empirical results suggest that the proposed routing mechanism can be beneficial. However, the contribution is somewhat incremental, the theoretical depth is limited, and the experimental results, although enhanced after revision, culd be improved further.